# Lanthanide single-atom catalysts for efficient CO₂-to-CO electroreduction

Qiyou Wang[1,2], Tao Luo[1,2], Xueying Cao[3], Yujie Gong[4], Yuxiang Liu[1], Yusen Xiao[1], Hongmei Li[1], Franz Gröbmeyer[5], Ying-Rui Lu [6], Ting-Shan Chan [6], Chao Ma [7], Kang Liu[1], Junwei Fu[1], Shiguo Zhang [7], Changxu Liu[8], Zhang Lin[2], Liyuan Chai[2], Emiliano Cortes [5] ✉ & Min Liu [1] ✉

Single-atom catalysts (SACs) have received increasing attention due to their 100% atomic utilization efficiency. The electrochemical CO₂ reduction reaction (CO₂RR) to CO using SAC offers a promising approach for CO₂ utilization, but achieving facile CO₂ adsorption and CO desorption remains challenging for traditional SACs. Instead of singling out specific atoms, we propose a strategy utilizing atoms from the entire lanthanide (Ln) group to facilitate the CO₂RR. Density functional theory calculations, operando spectroscopy, and X-ray absorption spectroscopy elucidate the bridging adsorption mechanism for a representative erbium (Er) single-atom catalyst. As a result, we realize a series of Ln SACs spanning 14 elements that exhibit CO Faradaic efficiencies exceeding 90%. The Er catalyst achieves a high turnover frequency of ~130,000 h⁻¹ at 500 mA cm⁻². Moreover, 34.7% full-cell energy efficiency and 70.4% single-pass CO₂ conversion efficiency are obtained at 200 mA cm⁻² with acidic electrolyte. This catalytic platform leverages the collective potential of the lanthanide group, introducing new possibilities for efficient CO₂-to-CO conversion and beyond through the exploration of unique bonding motifs in single-atom catalysts.

The escalating CO₂ emissions from fossil fuel consumption have exacerbated environmental crises like climate change and ocean acidification, highlighting the urgency to develop technologies for CO₂ capture and utilization[1,2]. The electrochemical CO₂ reduction reaction (CO₂RR) presents a promising strategy to convert CO₂ into value-added products like hydrocarbons and oxygenates[3]. Among the potential products, carbon monoxide (CO) holds particular importance as a vital feedstock for various chemical processes, including the production of methanol, acetic acid, and hydrocarbons via Fischer-Tropsch synthesis[4,5]. Presently, single-atom catalysts (SACs) with isolated active site have aroused widespread attention due to their prominent ability to inhibit *H combination within competitive hydrogen evolution[6–8]. A repertoire of atoms has been utilized as the catalysts, ranging from main group to transitional metals[7,9,10].

To improve the efficiency of SACs-based CO₂RR to CO, numerous strategies have been proposed to improve the weak *COOH adsorption with single-line pathway (Metal-C), focusing on the regulation of coordination number and atom species[11–15]. However, achieving strong *COOH adsorption on traditional metal atom is inherently difficult because of the weak Metal-C bonding by virtue of single-line

[1]Hunan Joint International Research Center for Carbon Dioxide Resource Utilization, State Key Laboratory of Powder Metallurgy, School of Physics, Central South University, Changsha, PR China. [2]School of Metallurgy and Environment, Central South University, Changsha, Hunan, PR China. [3]College of Materials Science and Engineering, Linyi University, Linyi, Shandong, PR China. [4]School of Electrical Engineering, University of South China, Hengyang, Hunan, PR China. [5]Nanoinstitute Munich, Faculty of Physics, Ludwig-Maximilians-Universität (LMU), Munich, Germany. [6]National Synchrotron Radiation Research Center, 300, Hsinchu, Taiwan. [7]College of Materials Science and Engineering, Hunan University, Changsha, PR China. [8]Centre for Metamaterial Research & Innovation, Department of Engineering, University of Exeter, Exeter, UK. ✉e-mail: Emiliano.Cortes@lmu.de; minliu@csu.edu.cn

pathway[16,17]. More importantly, strong *COOH adsorption may lead to difficult CO desorption because the binding strengths of these two intermediates via common Metal-C bond are positively correlated[18,19]. Thus, achieving facile *COOH adsorption and CO desorption simultaneously remains a significant challenge, resulting in a limited turnover frequency (TOF) in the range of few thousands to tens of thousands[11,12,20–23], and thus leading to low energy efficiencies and $CO_2$ conversion efficiencies.

Here, we introduce a approach to address this fundamental bottleneck during $CO_2RR$ to CO. Instead of relying on specific atoms to facilitate the *COOH adsorption and CO desorption, we unveil that the entire group of non-radioactive lanthanide (Ln) metals can serve as optimal catalysts for CO production from $CO_2RR$. The large atomic size of Ln metals, which is over twice the size of a carbon atom, significantly mitigates steric-hindrance effects for double-line (bridge) adsorption[24,25]. Meanwhile, the oxophilicity of Ln metals[26–28] facilitates the formation of strong bridge adsorption by pattern of Metal-C and Metal-O bonding for *COOH. Most importantly, the variable coordination numbers of Ln atoms endow their transition ability from *COOH bridge adsorption to *CO linear adsorption during $CO_2RR$ process. circumventing the scaling relationship between *COOH and *CO.

Despite variations in atomic number across the Ln series (from 57 to 71), we realized SACs incorporating fourteen Ln with CO faraday efficiencies exceeding 90%, highlighting the universal adaptability of this mechanism. Using erbium (Er) as a representative SAC, we elucidate the favorable *COOH bridge adsorption and CO desorption through density functional theory (DFT) calculations, operando attenuated total reflection infrared spectroscopy (ATR-IR), and Er $L_3$-edge XAS measurements. As a result, the flow cell fabricated with Er SAC exhibits a high turnover frequency (TOF) of approximately 130,000 h$^{-1}$ at 500 mA cm$^{-2}$, achieving a full-cell energy efficiency of 34.7% and single-pass carbon efficiency (SPCE) of 70.4% at 200 mA cm$^{-2}$ for CO production.

## Results and discussion
### DFT Calculations
To elucidate the improved adsorption/desorption on Ln single-atom catalysts (SACs) for $CO_2RR$, we conducted a systematic investigation using Density Functional Theory (DFT) calculations, comparing them with traditional SACs. For clarity without loss of generality, erbium (Er) was chosen as the representative Ln, while calcium (Ca) and iron (Fe) were selected as reference elements. Both Ca and Fe are employed as typical SACs for $CO_2RR$, with Fe known for its low energy barriers for $CO_2$ activation and Ca favoured for its facile CO desorption[9,10,29].

Schematically, the intermediate *COOH exhibits a bridge adsorption state on Er SAC, in contrast to the linear adsorption on Fe and Ca SACs (Fig. 1a, b and Supplementary Fig. 1). However, the intermediate *CO shares a linear adsorption on Er SAC, consistent with those on Fe and Ca SACs. Free energy diagram indicates that Er SAC (0.61 eV) and Fe SAC (0.63 eV) exhibit lower energy barriers for $CO_2$ activation than that of Ca SAC (1.46 eV), proving the bridge adsorption benefits *COOH formation (Fig. 1c). In addition, the CO desorption on Er SAC (0.34 eV) and Ca SAC (−0.07 eV) are easier than that on Fe SAC (1.06 eV). Thus, the bridge adsorption does not transition onto subsequent *CO linear adsorption on Er SAC, thereby circumventing the scaling relationship in terms of *COOH and *CO observed in typical SAC.

To further explore the interaction between metal sites and intermediates, we examined the catalyst-intermediate adduct (M-COOH, M-CO) (Fig. 1d and Supplementary Fig. 2). Charge density differences and Bader charge analysis are conducted to analyze the intensity of chemical bonds between intermediates and metal sites. As shown in Fig. 1d−f, show that charge transfer from Er and Fe SAC to *COOH is 0.500 and 0.584 e respectively, larger than that of Ca SAC (0.363 e), indicating strong *COOH adsorption on Er sites by virtue of bridge

adsorption. Furthermore, Er SAC (0.012 e) and Ca SAC (0.005 e) provide negligible charge to *CO, compared to Fe SAC (0.136 e), demonstrating the easier CO desorption on Er and Ca site than that on Fe site. Thus, Er SAC facilitates the $CO_2RR$ to CO pathway through the bridge adsorption of *COOH as well as the linear adsorption of CO.

### Catalyst synthesis and characterization
To utilize the unique catalytical properties predicted by DFT calculations, Er SAC was prepared on carbon nanotubes (CNTs) through a facile calcination method (Scheme in Supplementary Fig. 3). Only the XRD diffraction peaks of CNT substrate was found, suggesting the absence of metallic phases and a homogeneous dispersion of the Er atoms (Supplementary Fig. 4)[30,31]. No metal contamination was found in these precursors of Er SAC (Supplementary Figs. 5, 6). The metal contents in Er SAC were estimated to be ~2.17 wt% (Supplementary Table 1), according to inductively coupled plasma optical emission spectrometer (ICP-OES). The morphology and the atomic Er dispersion were investigated through different types of microscopies. The results from scanning electron microscope (SEM), high-resolution transmission electron microscope (HRTEM) and aberration-corrected high-angle annular dark-field scanning transmission electron microscopy (AC HAADF-STEM) confirmed the single-atom nature of the Er sites on CNT (Fig. 2a and Supplementary Fig. 8). In addition to Er, we developed a systematic procedure to SACs supported on CNTs utilizing all Ln expect radioactive Promethium (Pm), encompassing Lanthanum (La), Cerium (Ce), Praseodymium (Pr), Neodymium (Nd), Samarium (Sm), Europium (Eu), Gadolinium (Gd), Terbium (Tb), Dysprosium (Dy), Holmium (Ho), Thulium (Tm), Ytterbium (Yb), and Lutetium (Lu). Significantly, the size of Ln atom is much larger than that of Ca, and Fe atoms. Further investigations show that 16 SACs including Ln, Ca, and Fe SACs have similar components and morphologies as those of Er SAC (Supplementary Figs. 7–25), demonstrating the universality of preparation.

To acquire the structural information of SACs, synchrotron-based X-ray adsorption near edge structure (XANES) spectra were conducted. Er SAC shows a clear increase of C K-edge peak intensity at ~288.5 eV, suggesting the possible formation of C-N-Er bonds, compared to nitrogen doped carbon nanotubes (NC) (Fig. 2c)[32–34]. The increase of pyridinic N peak at 400.7 eV indicates Er atoms dominantly coordinate with pyridinic N atoms in Er SAC (Fig. 2d)[35,36]. Fourier transformed (FT) extended X-ray adsorption fine structure (EXAFS) manifests the atomic dispersion features of the Er, Fe and Ca atoms, with a coordination number of ~6, ~4 and ~4 based on well-fitting process respectively, consistent with the results from theoretical calculations (Fig. 2e−g and Supplementary Figs. 10, 11, Supplementary Table. 2).

To study the real-time intermediates formed on metal sites during $CO_2RR$ to CO, we carried out operando attenuated total reflection infrared spectra (ATR-IR) and $L_3$-edge XAS measurements for Er (Supplementary Figs. 26, 27). Peaks located at range from 1910 to 1950 cm$^{-1}$ and around 1640 cm$^{-1}$ can be attributed to *CO (linear-bonded CO) and $H_2O$ bending respectively (Fig. 3a, b and Supplementary Fig. 28 and Supplementary Table 3)[37,38]. Notably, a new peak ranging from 1800 to 1840 cm$^{-1}$ was identified, assigned to bridge adsorption of *COOH on Er SAC, distinguishing it from Fe and Ca SAC[9,39,40]. Similarly, Er and Fe SACs show stronger $CO_2$ adsorption signals of $CO_2$ adsorption based on temperature program desorption, compared with that of Ca SAC, facilitating the subsequent $CO_2$ activation on Ca and Fe sites (Supplementary Fig. 29). Furthermore, differing with the distinct CO adsorption peaks on Fe SAC, no such peak was observed on Er and Ca SACs, suggesting facile CO desorption from Er and Ca sites, consistent with the results from DFT calculations.

We further investigated the chemical structure and coordinating environment of Er SAC under operating conditions using XAS. The intensity of white line heightened slightly due to the increased

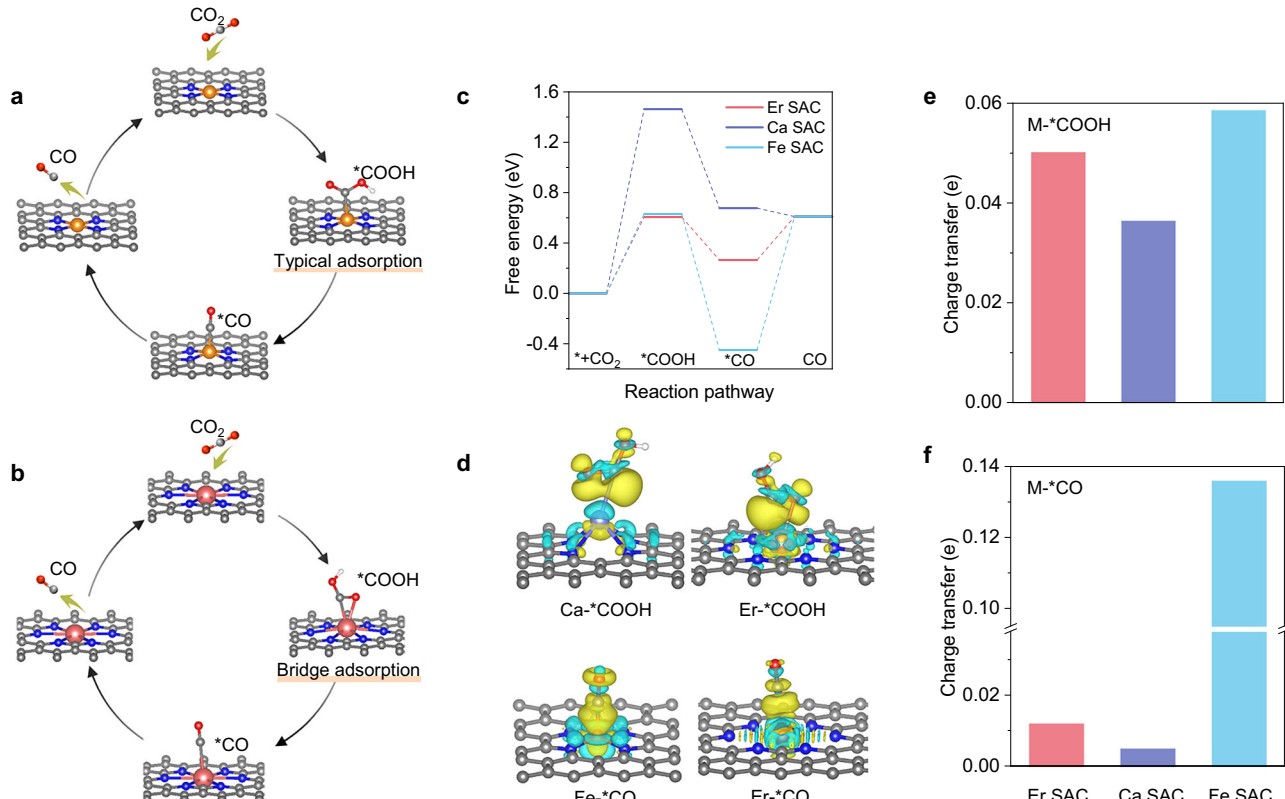

**Fig. 1 | Theoretical calculations.** Structure and adsorption configurations of key intermediates on Fe SAC (**a**) and Er SAC (**b**). **c** Free energy diagram for $CO_2$ reduction to CO. **d** Charge density differences for *COOH and *CO adsorbed on SAC. Yellow, charge density accumulation; green, charge density depletion. Bader charge transfer analysis for COOH adsorbed on SAC (**e**) and CO adsorbed on Metal (M) SAC (**f**). Atom color-coding: Red, oxygen; white, hydrogen; grey, carbon; blue, nitrogen; orange, Fe; pink, Er.

chemical state during $CO_2$ electroreduction (Fig. 3c). The Er-N bond position of Er SAC shifted towards the Er-O side during $CO_2$RR, demonstrating the existence of Er-O from Er-*COOH bridge adsorption[41,42]. Importantly, over the applied potential range relevant to $CO_2$RR, the Er-Er bond was not observed (Fig. 3d), suggesting that Er SAC maintained its original complex status without reduction to Er nanoparticles or nanoclusters under operating conditions[39,43].

### Evaluating catalyst performance for $CO_2$RR

To evaluate the performance of catalysts, electrochemical tests were first conducted in $CO_2$ saturated 0.5 M $KHCO_3$ electrolyte (Supplementary Figs. 30–40). Er SAC shows high Faradaic efficiencies of CO ($FE_{CO}$) ≥90% over a wide potential range from −0.47 to −0.97 V vs. RHE (Fig. 4a), with a maximal $FE_{CO}$ reaching ~99% (Supplementary Fig. 38). Furthermore, Er SAC retains a $FE_{CO}$ above 90% at −0.67 to −0.87 V vs. RHE, even when the $CO_2$ concentration was reduced to 30% (Supplementary Fig. 36). The flow cell fabricated with Er SAC shows a large current density of 500 mA $cm^{-2}$ under high CO Faradaic efficiency ≥90% in 1 M $KHCO_3$ electrolyte (Fig. 4b and Supplementary Figs. 41, 42).

To avert low carbon utilization limits witnessed in neutral solutions[44–46], the performance of Er SAC was tested in acidic media, 1 M KCl (pH adjusted to 1.0 with sulfuric acid). Only tiny degradation of $FE_{CO}$ was observed when then current densities were increased by 10 times from 50 to 500 mA $cm^{-2}$, demonstrating its potential for large-scale production. Such ≥90% Faradaic efficiency at 500 mA $cm^{-2}$ both in neutral (top panel, Fig. 4b) and acidic electrolyte (bottom panel, Fig. 4b and Supplementary Fig. 43) endows the SAC a high turnover frequency (TOF) of ~130,000 $h^{-1}$. Furthermore, Er SAC can achieve a full-cell energy efficiency of 34.7% at 200 mA $cm^{-2}$ (Supplementary Figs. 44–47 and Supplementary Tables 4, 5)[47–50]. To decrease resistance

of the system, we have tested the performance in the Membrane Electrode Assembly (MEA). As shown in Supplementary Table 5, MEA fabricated Er SAC shows high full-cell energy efficiency of ~32.5% even at high current density of 300 mA $cm^{-2}$.

Moreover, we investigated the catalytic performance across various $CO_2$ flow rates spanning a wide range of current densities. Notably, we achieved a remarkable single-pass carbon efficiency (SPCE) of 70.4% for $CO_2$RR to CO a large current density of 200 mA $cm^{-2}$, indicating that 70.4 out of 100 $CO_2$ molecules can be successfully transformed into CO at the outlet (Fig. 4c, d and Supplementary Tables 6–11). The flow cell maintained stable operation at 100 mA $cm^{-2}$, with $FE_{CO}$ >90% in acidic electrolyte (100 h, Fig. 4e), and meanwhile showed a long-term stability in neutral electrolyte (Supplementary Fig. 45). No obvious cluster and Er-Er bond are observed on Er SAC after the test according to operando XANES spectra (Fig. 3d) and TEM images, demonstrating the durability of structure (Supplementary Fig. 46).

Figure 4f illustrates a comparative analysis of catalytic performance between our study and a previous flow cell for $CO_2$ to CO conversion. To provide a comprehensive evaluation, we considered five performance indicators: efficiency (both FE and SPCE), stability (duration for $FE_{CO}$ exceeding 90%), pH, and the operational current at maximum SPCE. Our Er SAC surpassed previous reported data across all performance metrics, a testament to the efficacy of the unique mechanism elucidated earlier (Supplementary Tables 12–14).

To assess the generic catalytic abilities of Ln, similar tests were conducted for the other thirteen SACs utilizing different elements. All of Ln SACs keep a high $FE_{CO}$ between 92.0% and 99.6% at −0.67 V vs. RHE (Fig. 4a). The consistently high efficiency observed across SACs using elements from the entire Ln group (except radioactive Pm)

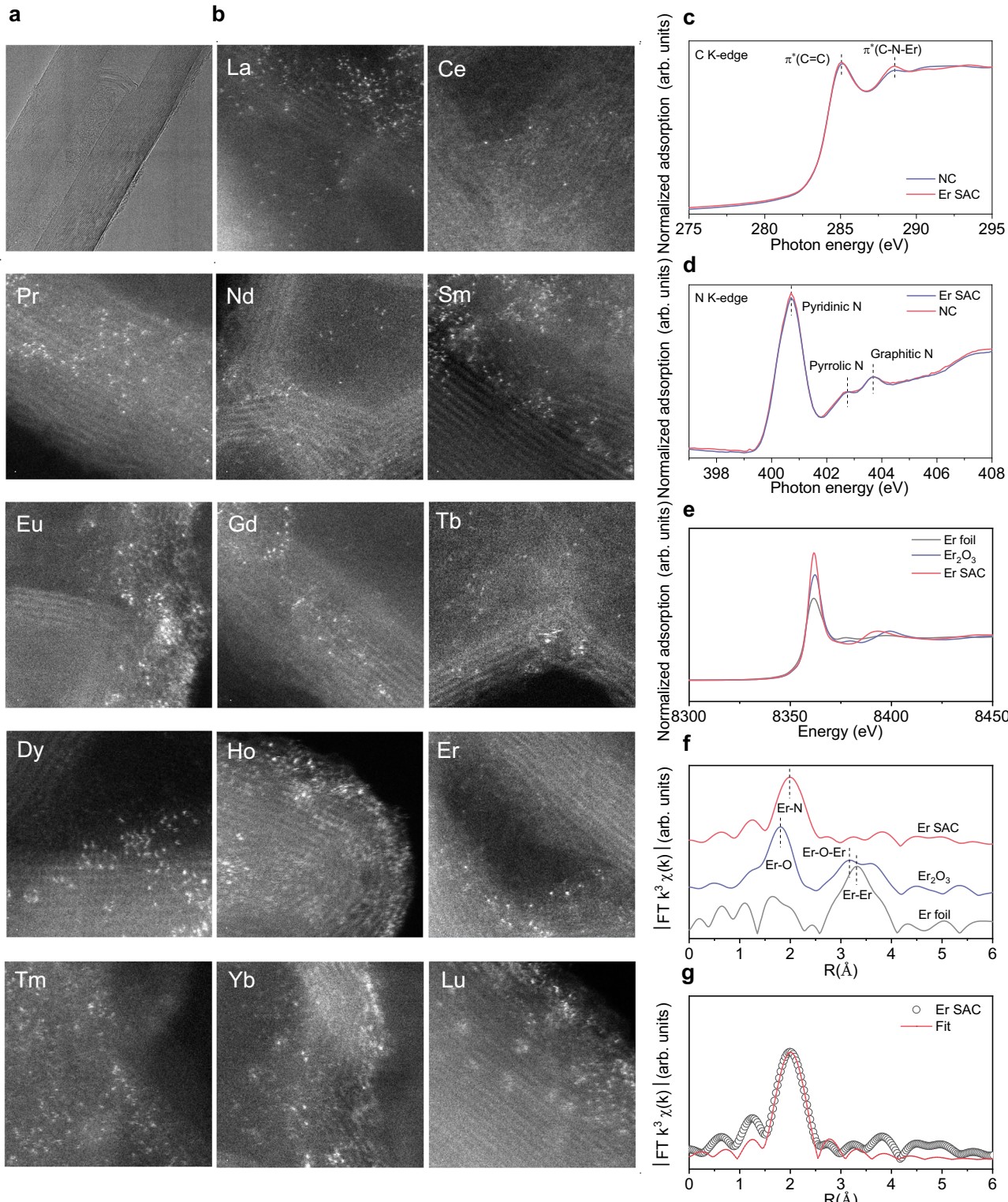

**Fig. 2 | Structural characterization. a** HRTEM image for Er SAC. **b** AC HAADF-STEM images of Ln SACs. Scale bar, 2 nm. **c** XANES spectra of C K-edge. **d** XANES spectra of N K-edge. **e** Er $L_3$-edge XANES spectra of Er foil, $Er_2O_3$ and Er SAC. **f** $k^3$ weighted Fourier transform spectra of Er foil, $Er_2O_3$ and Er SAC. **g** the corresponding EXAFS R space fitting curves for Er SAC.

unequivocally demonstrates the universal catalytic activity of Ln as predicted by DFT and indicated by operando spectra.

In summary, we proposed and realized a series of Ln SACs with catalytic performance for $CO_2RR$ to CO. The unique electron configuration shared by Ln metals facilitates the bridge adsorption for favorable *COOH formation, which circumvents the scaling relationship between

*COOH and *CO. Both DFT calculations and experimental characterizations were employed to prove the bridge adsorption of *COOH for Er SAC.

The adaptability of our strategy was validated through $CO_2RR$ flow cells containing SACs incorporating 14 different Ln metal atoms, all exhibiting $FE_{CO}$ beyond 90%. Among them, the Er SAC demonstrated a remarkable TOF of ~130,000 $h^{-1}$ at a high current density of

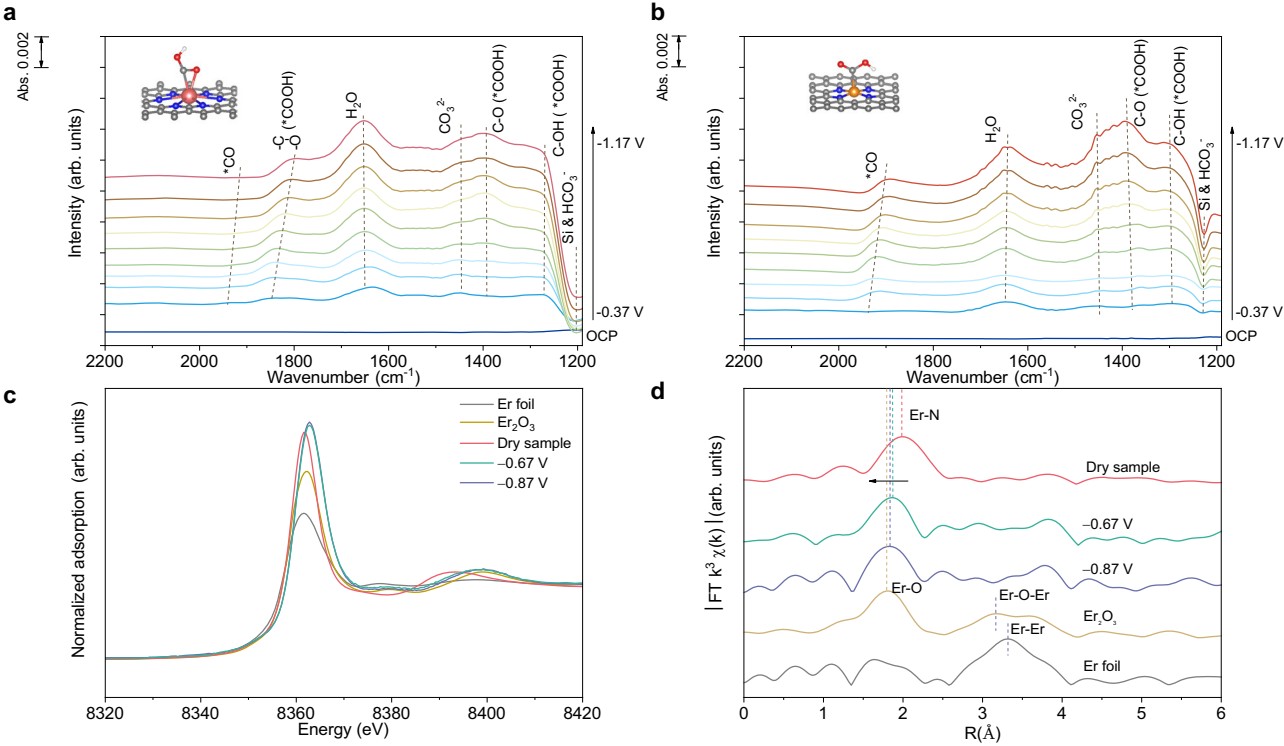

**Fig. 3 | Operando spectra analysis. a** operando ATR-IR spectra of Er SAC (without iR-correction). **b** operando ATR-IR spectra of Fe SAC (without iR-correction). **c** operando XANES spectra of Er SAC measured at different potentials (without iR-correction). **d** Corresponding operando k³ weighted Fourier transform EXAFS spectra of Er SAC (without iR-correction). Atom color-coding: Red, oxygen; white, hydrogen; grey, carbon; blue, nitrogen; orange, Fe; pink, Er.

500 mA cm$^{-2}$, maintaining a CO Faradaic efficiency of ≥90% in both neutral and acidic electrolytes. This achievement results in a significant enhancement of full-cell energy efficiency to 34.7% and single-pass CO$_2$ electrolysis (SPCE) to 70.4% at a large current density of 200 mA cm$^{-2}$. These results underscore the promising potential of our approach for practical industrial applications. Furthermore, with the recent pioneering work involving promethium (Pm), the investigation of this only missing lanthanide element in our configurations will be an interesting and valuable direction for future research[51].

While previous studies have primarily focused on identifying specific Ln atoms for targeted chemical transformations such as nitrogen reduction, carbon dioxide reduction, and oxygen reduction reactions[31,52-55], our approach represents a departure from this paradigm. Rather than singling out individual atoms, we present a methodology that harnesses the collective catalytic potential of the entire Ln group. The significance of our work extends beyond the demonstrated conversion of CO$_2$ to CO; it opens avenues for exploring the applicability of our strategy (using the entire Ln group) to a broader range of reduction reactions and beyond. In addition to its fundamental implications, our approach offers an platform for CO$_2$ neutralization, with the unique ability to select different Ln atoms without compromising efficiency. This flexibility introduces potential benefits for practical applications, where the choice of Ln metals can be guided by considerations such as availability, cost, and compatibility, in addition to catalytic properties.

## Methods
### Preparation of single atom catalysts
La(NO$_3$)$_3$·6H$_2$O (99.9%), Ce(NO$_3$)$_3$·6H$_2$O (99.9%), Pr(NO$_3$)$_3$·6H$_2$O (99.99%), Nd(NO$_3$)$_3$·6H$_2$O (99.9%), Sm(NO$_3$)$_3$·6H$_2$O (99.9%), Eu(NO$_3$)$_3$·6H$_2$O (99.9%), Gd(NO$_3$)$_3$·6H$_2$O (99.9%), TbCl$_3$·6H$_2$O (99.9%), Dy(NO$_3$)$_3$·6H$_2$O (99.9%), HoCl$_3$·6H$_2$O (99.9%), Er(NO$_3$)$_3$·6H$_2$O (99.9%), TmCl$_3$·6H$_2$O (99.9%),

YbCl$_3$·6H$_2$O (99.9%), LuCl$_3$·6H$_2$O (98%), FeCl$_3$ (99.9%), CaCl$_2$ (99.9%), NaCl (99.9%), KCl (99.9%) and Dicyandiamide (99.9%) were bought from Shanghai Aladdin reagent co. Ltd. Carboxylated carbon nanotube (CNT, 30-50 nm in diameter) and Ketjen black were purchased from Pioneer Nanotechnology Co. Ltd. Before using CNT, 0.5 M HNO$_3$ (98%) was used to remove the potential metal impurities at 80 °C for 12 hours.

Synthesis of single atom catalysts (SACs) followed a two-step procedure. The first step synthesis of C$_3$N$_4$ nanosheets (NS), Briefly, 0.88 g NaCl, 1.12 g KCl and 6 g dicyandiamide were grinded, followed by being heated in a muffle furnace at 670 °C with a 2 °C/min heating rate and then retained 670 °C for 45 min. The mixture after pyrolysis was dissolved in 200 mL deionized (DI) water under ultrasound treatment and 100 mL ethanol was added to precipitate C$_3$N$_4$ NS out. Then centrifugation was employed to remove liquid supernatant. For further purification, the mixture was transferred to a dialysis bag (MD55-3500) and dialysis in DI water lasted more than 7 days. The C$_3$N$_4$ NS was obtained by rotary evaporation at 60 °C. The second step is the preparation of SACs. Briefly, 20 mg C$_3$N$_4$ NS and 60 mg CNT were spread out in 30 mL DI water and shattered by 60 min ultrasound. 0.05 mL of 0.1 M of the corresponding metal salt (nitrate or chloride) was added slowly followed by stirring for 2 h. The liquid nitrogen was added directly into the mixture to obtain an ice block, which was then freeze-dried for 72 h to acquire aerogel. The aerogel was pyrolyzed under the condition of 730 °C with a 5 °C/min heating rate at Ar atmosphere, without any heat preservation, and cooled to 25 °C. Note that, to inhibit the oxidation toward O coordination environment, the Ca SAC should be vacuum sealed before characterization and electrochemical test[10].

### Characterizations
The X-ray diffraction (XRD) patterns were collected through using a D8 advance X-ray diffractometer (Rigaku, Japan) with Cu Kα

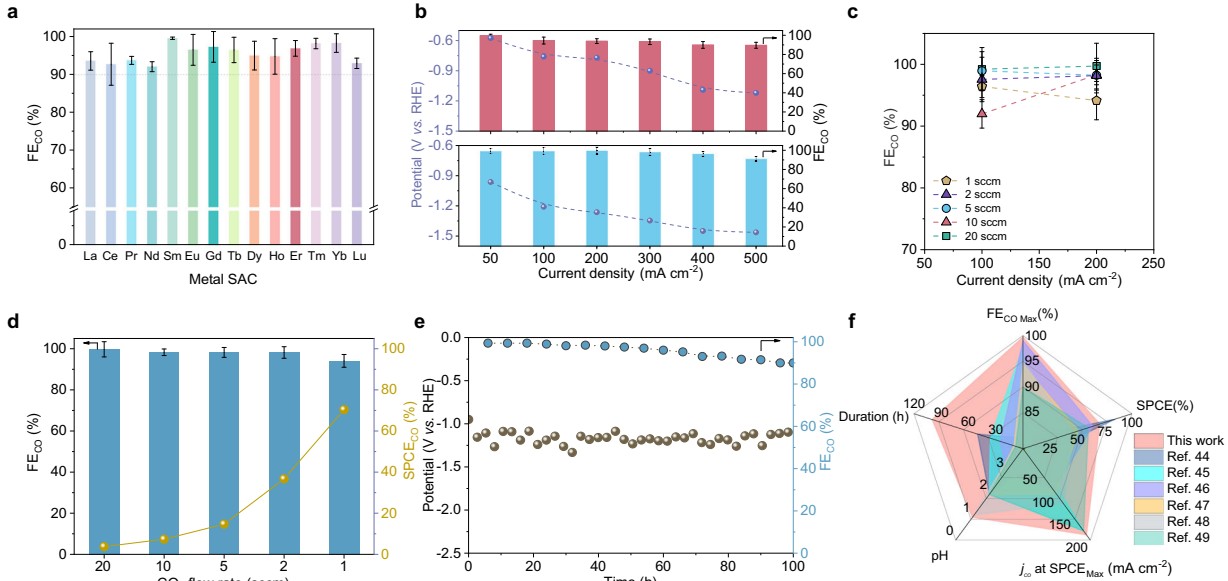

**Fig. 4 | Electrochemical CO₂RR performances. a** $FE_{CO}$ for Ln SACs at −0.67 V vs. RHE (without iR-correction) in H-cell under $CO_2$ saturated 0.5 M $KHCO_3$ solution. **b** Potentials and $J_{CO}$ for Er SAC at different current densities with 1 M $KHCO_3$ solution (upper) and 1 M KCl (pH = 1, down) in flow cell (95% iR-correction, Resistance = $2.4 \pm 0.3 \, \Omega$ and $1.8 \pm 0.2 \, \Omega$ for neutral and acidic electrolyte, respectively). **c** $FE_{CO}$ and current densities for Er SAC at different $CO_2$ flow rates at 1 M KCl (pH = 1). **d** $FE_{CO}$ and SPCE of $CO_2$ to CO on at different $CO_2$ flow rates (pH=1, applied current density of −200 mA cm⁻²). **e** Stability of Er SAC at a current density of −100 mA cm⁻² with (pH = 1, potential with 95% iR-correction). **f** Comparison of $CO_2RR$-to-CO performance with the state-of-art electrocatalysts in acidic media[68–73]. The error bars correspond to the standard deviations of measurements over three separately prepared samples under the same testing conditions.

radiation (λ = 0.15406 nm) at a scan rate (2θ) of 5 °/min. The morphologies of the samples were imaged by field emission scanning electron microscopy (SEM, Hitachi S-4800) and high-resolution transmission electron microscopy with a spherical aberration corrector (HRTEM, Titan G2 60-300) equipped with X-ray energy dispersive spectrometer (EDS). The atomically dispersed metal atoms were detected by aberration-corrected HAADF-STEM (Thermo Scientific, Themis Z). The BET specific surface areas were obtained from JW-BK200C nitrogen sorption analyzer (Beijing JWGB SCI. & Tech. Co., Ltd) with 100 °C pre-treatment in high vacuum, and the pore size distribution was calculated from the adsorption branch of the isotherms. X-ray photoelectron spectroscopy (XPS) measurements were performed on Thermo Fisher Scientific Escalab 250 XI, and all the binding energies were calibrated by the C 1s peak at 284.8 eV. C and N k-edge X-ray adsorption spectra were collected from the BL12B-a beamline in National Synchrotron Radiation Laboratory (NSRL) and BL08U1A beamline of Shanghai Synchrotron Radiation Facility (SSRF). The photon flux was $1 \times 10^{10}$ photons per second. The energy of the adsorption spectrum was first calibrated with the Au 4 f peak. Lanthanide, Fe K-edge, Ca K-edge and Lanthanide L₃-edge of X-ray adsorption spectra were obtained at beamlines 01C1 and 16A1 in the National Synchrotron Radiation Research Center (NSRRC, Taiwan). Raman spectra were obtained by a DXRI Raman Microscope (Thermo Fisher) using a 532 nm laser as the light source. $CO_2$ temperature program desorption (TPD) curves were measured on Micromeritics AutoChem 2920. The content of metal atoms in the samples were measured with inductively coupled plasma optical emission spectrometer (ICP-MS, Agilent 7700). The gas phase products from electrolysis were quantified by the on-line Gas chromatograph (GC, Shimidzu, Model 2014).

### Electrochemical measurements
All the electrochemical measurements were operated with an electrochemical station and at 25 °C. Constant potential electrolysis was carried out at various potentials for 60 min to analyze the products.

The H-cell with two 30 mL chambers were separated by an anion exchange membrane (Selemion DSVN, 95 μm of thickness). Working electrode is made up of carbon paper (Toray, TGP-H060) with catalysts coating, Ag/AgCl reference electrode (3.5 M KCl) and Pt mesh counter electrode (1*1 cm²). At the high-purity hydrogen saturated electrolyte, using Pt foil as working and counter electrode to calibrate Ag/AgCl reference electrode. Potentials for LSV curves in H-cell were referenced to reversible hydrogen electrode (RHE) with the formula of E (RHE) = E (Ag/AgCl) + 0.205 V + 0.059 V × pH after iR compensation. The uncompensated solution resistance was compensated for 95% by EIS measurement. Potentials for constant potential electrolysis were not compensated. The preparation of electrode was as follows: 5 mg catalyst was added into 970 μL isopropanol and 30 μL Nafion solutions (5 wt%, Sigma-Aldrich), followed by sonication of 30 min to form a homogeneous catalyst ink. The catalyst ink was dropped onto carbon paper (0.25 cm²) directly and then dried at 60 °C for 12 hours, which ensured 0.2 mg cm⁻² mass loading of the catalyst. The electrolyte was $CO_2$ (20 sccm, calibrated by mass flow controller) saturated 0.5 M $KHCO_3$ (pH = 7.28 ± 0.05).

The flow cell comprises gas diffusion electrode (GDE, SGL29BC), sandwich of flow frames, gaskets, counter electrode, reference electrode and exchange membrane (95 μm-thickness Selemion DSVN for $KHCO_3$ electrolyte; 117 μm-thickness Nafion 117 for acidic electrolyte). No special treatment before using Selemion DSVN. Before using, Nafion 117 was treated with 5% hydrogen peroxide at 80 °C for 1 hour, then boiled in 5% dilute sulfuric acid at 80 °C for 1 hour. The preparation of working electrode was home-made. Briefly, 1.5 mg catalyst was added into 950 μL isopropanol, 75 μL PTFE solutions (Polytetrafluoroethylene, 3 wt%) and 24 μL Nafion mixed solutions (5 wt%, Sigma-Aldrich), followed by sonication of 30 min to form a homogenous catalyst ink. The catalyst ink was sprayed on a hydrophobic GDE (3 cm²) and then dried at 70 °C for 8 h. The loading of the catalyst is 0.5 mg cm⁻² and the area contacting electrolyte is 0.5 cm². The $IrO_2$-coating titanium

foil is employed as counter electrode and an Ag/AgCl (with saturated 3.5 M KCl) electrode as reference electrode. The flow rate of the electrolyte was set at 30 mL min$^{-1}$ in both cathodic and anodic chambers. The cathodic acidic electrolyte and anodic acidic electrolyte in flow cell are 1 M KCl (pH adjusted to 1.02 ± 0.02, with sulfuric acid) and 0.5 $H_2SO_4$ (0.01 ± 0.01) respectively. Before the test, all the electrolyte is stored in the refrigerator. The cathodic neutral electrolyte and anodic neutral electrolyte in flow cell are both 1 M $KHCO_3$. Potentials for flow cell were referenced to reversible hydrogen electrode (RHE) with the formula of E (RHE) = E (Ag/AgCl) + 0.205 V + 0.059 V × pH after $iR$ compensation. The uncompensated solution resistance was compensated for 95% by EIS measurement.

The Membrane Electrode Assembly (MEA) fabrication is like that of flow cell, except no reference electrode and cathodic electrolyte chamber. The area contacting electrolyte in MEA is also 0.5 cm$^2$, and the electrolyte is 0.5 M $K_2SO_4$ (pH adjusted to 1.0 with sulfuric acid) avoiding chlorine evolution on anode.

The cathodic products were analyzed by an on-line gas chromatograph. High-purity $N_2$ (99.999%) was used as the carrier gas. A TCD was employed to measure the $H_2$ fraction, and a flame ionization detector equipped with a nickel conversion furnace was used to analyze the CO fraction. Mass flowmeters of different ranges are used to measure the $CO_2$ flow rate and calibrated by soap film flowmeter. The Faradaic efficiency of products was assessed from gas chromatogram peak according to the flowing equation:

$$FE_{CO\,orH_2} = x \times V \times \frac{2FP_0}{iRT} \quad (1)$$

x: fraction value V: flow rate of $CO_2$ F: faraday constant (96485 C/mol) $P_0$: normal atmosphere (101325 Pa), I: applied current R: gas constant (8.314 J/(mol·K)) T: room temperature (298 K, 25 °C).

Noticeably, it took more than one hour and two hours of electroreduction time to wait for the gas components balance in the chamber when testing the faradaic efficiency at 2 and 1 sccm flow rates respectively. The GC standard curve was calibrated by using standard mixture gas with different concentrations, especially at high-concentration products at low flow rates.

### TOF calculations

We calculate the TOF according to the following equation:

$$TOF(h^{-1}) = \frac{I_{product}/nF}{m_{cat} \times \alpha/M_{metal}} *3600 \quad (2)$$

$I_{product}$: partial current for CO, An: number of electrons transferred for CO, 2F: Faradaic constant, 96,485 C/mol $m_{cat}$: catalyst mass in the electrode, g α: mass ratio of active atoms in catalysts $M_{metal}$: atomic mass of metal

**Full-cell energy efficiency (EE) calculations:**

$$EE(\%) = 100\% \times \frac{E_a - E_c}{V_{full}} \times FE_{CO} \quad (3)$$

where $V_{full}$ and $FE_{CO}$ denote the full-cell voltage and FE of CO, respectively. $E_a$ and $E_c$ are the standard reduction potentials for the anode and cathode ($CO_2$-to-CO) reactions, respectively[50].

**SPC of $CO_2$ calculations at 25 °C, 1 atm:**

$$CO_{2\,consumed}\,(L\,min^{-1}) = (j\,mA\,cm^{-2}) \left(\frac{1\,A}{1000\,mA}\right) \times \left(\frac{60\,s}{1\,min}\right) \times \left(\frac{1\,mol\,e^-}{96485\,C}\right)$$
$$\times \left(\frac{1\,mol\,CO}{2\,mol\,e^-}\right) \times \left(\frac{1\,mol\,CO_2}{1\,mol\,CO}\right) \times \frac{24.05\,L}{1\,mol\,CO_2} \times (0.5\,cm^2) \quad (4)$$

$$SPC(\%) = 100\% \times \left(\frac{CO_{2\,consumed}\,(L\,min^{-1})}{CO_2\,flow\,rate\,(L\,min^{-1})}\right) \quad (5)$$

Where j is the partial current density CO production from $CO_2$ reduction[56,57].

### Operando attenuated total reflection-infrared spectroscopy (ATR-IR)

ATR-IR was conducted on a Nicolet iS50 FT-IR spectrometer. The Au-coated Si semi-cylindrical prism (20 mm in diameter) was employed as the conductive substrate for catalysts and the IR refection element. The catalyst's ink was dropped on the Au/Si surface as the working electrode and the mass loading of the catalyst was 0.5 mg/cm$^2$. Operando ATR-IR spectra were recorded during stepping the working electrode potential[39].

### Computational methods

Density functional theory (DFT) calculations were employed by Vienna Ab initio Simulation Package (VASP) with the projector augment wave (PAW) method[58–62]. The exchange and correlation potentials were present in the generalized gradient approximation with the Perdewe-Burkee-Ernzerh of (GGA-PBE)[63]. A vacuum region of 15 Å is employed to decouple the periodic replicas to avoid the interaction in-fluence of the periodic boundary conditions. Spin polarization was considered in all calculations. van der Waals (VDW) interactions were corrected using the D3 method of Grimme[64]. Meanwhile, a k-point Γ-centered mesh is generated for Brillouin zone samples for geometry optimization. The energy cutoff, convergence criteria for energy and force were set as 500 eV, 10$^{-5}$ eV/atom and 0.01 eV/Å, respectively. The optimized computational models are provided in supplementary data 1.

The computational hydrogen electrode (CHE) model was used to calculate the free energy diagram[65–67]. The reaction free energy (ΔG) was calculated as follows:

$$\Delta G = \Delta E + \Delta ZPE - T*\Delta S \quad (6)$$

where ΔE is the chemisorption energy calculated by the DFT method. ΔZPE and ΔS are the differences in zero-point energies and entropy during the reaction, respectively.

## Data availability

Full data supporting the findings of this study are available within the article and its Supplementary Information, as well as from the corresponding author upon reasonable request. Source data are provided with this paper.

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

## Acknowledgements

We gratefully thank the Foundation for Innovative Research Groups of the National Natural Science Foundation of China (No. 52121004 to M.L.), Natural Science Foundation of China (Grant No. 22002189, 22376222, 52372253, 52202125 to M.L. and J.F.), Science and Technology Innovation Program of Hunan Province (Grant No. 2023RC1012), Central South University Research Programme of Advanced Interdisciplinary Studies (Grant No. 2023QYJC012 to J.F.), Central South University Innovation-Driven Research Programme (Grant No. 2023CXQD042 to M.L.). We also acknowledge funding and support from the Deutsche Forschungsgemeinschaft (DFG) under Germany´s Excellence Strategy – EXC 2089/1 – 390776260 cluster of excellence e-conversion to E.C., the Bavarian program Solar Technologies Go Hybrid (SolTech), the Center for NanoScience (CeNS) and the European Commission through the ERC PoC SURFLIGHT to E.C. We would like to acknowledge the help from the TLS 01C1 and 16A1 beam lines of National Synchrotron Radiation Research Center (NSRRC, Taiwan) for various synchrotron-based measurements. We gratefully thank the BL08U1A beam line of Synchrotron Radiation Facility (Shanghai) and the BL10B beam line of National Synchrotron Radiation Laboratory (Hefei). We are grateful for resources from the High-Performance Computing Center of Central South University.

## Author contributions

M.L., E.C., K.L., and J.F. conceived, supervised and financed the project. Q.W., X.C., Y.X., F.G., C.L., Z.L., and L.C. designed the experiments and analyzed the results. Q.W., X.C., Y.G., Y.L. (Yuxiang Liu) and H.L. synthesized the samples, performed the electrochemical experiments, and analyzed the results. T.L. carried out the simulations and wrote the corresponding section. Y.L. (Ying-Rui Lu) and T.C. conducted the XAS measurements. C.M. and S.Z. carried out electron microscope measurements. All authors participated in the writing, read and commented on the manuscript.

## Funding

## Competing interests

The authors declare no competing interests.
