## [Transparent Peer Review file · Nature Communications]

Lanthanide Single-Atom Catalysts for Efficient CO₂-to-CO Electroreduction

Corresponding Author: Professor Emiliano Cortes

Version 0:

Reviewer comments:

Reviewer #1

(Remarks to the Author)

This article reports a group of Ln MNC catalysts (up to 14 catalysts) for CO₂ electroreduction to CO, with all selective for CO production. While some of them have been previously, refs, 31,58-61, the authors claim that, a key feature of these catalysts is the bridge adsorption of *COOH, which is different from common MNC with metals with low atomic number, such as Fe and Ca. If this were novel, the authors should validate their hypothesis with convincing experimental evidences, rather than only showing DFT results. Generally, this bridge adsorption is hardly stable, even from theoretical perspective. Furthermore, the presentation of the performance is very questionable (specific comments below). The CO₂ reduction performance is routine for MNC catalysts and indeed one could not see their advantages compared to common MNC catalysts such as Ni-N-C. Overall, the current manuscript is not suitable for publication in Nature Communications.

1. The bridge adsorption of *COOH, with both C and O bound with Ln metal atoms, is highlighted. My major concern, is such an adsorption mode really stable? It is rationalized by the authors by the large size of Ln atoms. But, such a bridge adsorption mode on larger metal atoms MNC such as Ir, Pt, Au have never been reported. If this is the case, why do not these MNC show similar performance? Convincing evidences on this key point should be provided.

2. *CO is still adsorbed linearly on Er-N-C as common metals. How does the transition from bridge adsorption to linear adsorption occur?

3. The Er-N coordination number is 6. This is also unusual for MNC catalysts.

4. Why is the IR peak at 1800 to 1840 cm⁻¹ assigned to bridge adsorption of *COOH? Can it be bridge adsorbed CO? The identification of the presence of the bridge-adsorbed *COOH intermediates is the most important point.

5. Performance presentation

1). The statement "Such $\geq 90\%$ Faradaic efficiency at 500 mA/cm² both in neutral (top panel, Fig. 4b) and acidic electrolyte (bottom panel, Fig. 4b and Supplementary Fig. 43) endows the SAC a high turnover frequency (TOF) of $\sim 130,000$ h⁻¹, simultaneously representing an unprecedented full-cell energy efficiency of 42.2% (Supplementary Fig. 44-46 and table 4)" is quite confusing. At which current density, potential and electrolyte the TOF was calculated? And, more important, the so-called high full cell energy efficiency of 42.2% is actually obtained at 50 mA/cm² (Supplementary Fig. 44). All the performance metrics should be presented in a clear way. In Supplementary Table 4, the corresponding current densities should be clearly provided. On Page 15, full-cell voltage (V_{full}) was used in the calculation of energy efficiency. How was the full-cell voltage measured in a flow cell with three-electrode system?

2). It is important to indicate at which current density the energy efficiency and carbon efficiency were obtained, in Supplementary Table 6.

3). It is also important to indicate with which kind of cell the performance was evaluated, in Supplementary Table 7, as the H-cell performance is for sure much lower than flow cell performance for a given catalyst. Obtaining an ultrahigh TOF is routine, using gas diffusion electrode with a low catalyst loading, especially for an element with high atom number.

4). NiNC is generally considered the most active among MNC catalysts with common metal elements. Is the ErNC better than NiNC? Please show such a performance comparison in a fair way, also with current density and FE.

Reviewer #2

(Remarks to the Author)

The manuscript "Lanthanide Single-Atom Catalysts for Efficient CO₂-to-CO Electroreduction" by Q. Wang et al provides a comprehensive study of the synthesis and evaluation of lanthanide group single atom catalysts for CO₂RR, especially elucidating the bridging adsorption mechanism. Also, the authors applied a vast variety of methods to study the materials

including theoretical calculation, physical characterization as well as electrochemical analysis. I recommend that this work can be published in Nature Communications after addressing the following minor issues:

1. Fe SAC shows a low CO selectivity in electrochemical CO₂ reduction especially at -0.57 V. Moreover, since the bubbles from HER reaction would weaken the signals of *CO, it should be difficult to observe the peak of *CO. However, an obvious peak ascribed to the adsorption of linear *CO on the Fe site could be observed in the operando ATR-IR spectra.
2. The pure CNTs should be analyzed in detail. Since the contents of single metal atom is very low and it indeed influences the performance of CO₂RR, the impurities of the raw materials should be eliminated thoroughly. Some detection such as thermogravimetric analysis should be operated to support the conclusion.
3. The FT-IR spectra of Er SAC should be provided to exclude the existence of C₃N₄. and the Er₂O₃ PDF of XRD in supporting information to prove the results regarding the feature of atomic dispersion.
4. In this work, Er SAC achieves durability at -100 mA/cm² for 100 hours. Why did this performance of flow cell have drop after 100 h. Is there a destruction in the Er single atom catalyst, such as dissolution or aggregation? Or did the catalyst break off and flooding occur during the flow-cell test? The author should find the reason carefully.
5. Some minor errors should be corrected carefully, such as the format in formula 4, 5, The titles of horizontal axis in Figure 2f and 3d, and the useless line in Figure S3.

Reviewer #3

(Remarks to the Author)

This manuscript reports the universal properties of Lanthanide SACs for CO production via the electrochemical reduction of carbon dioxide. The underlying principle is intriguing and is expected to attract considerable interest from readers in the field of catalysis. However, several issues still need improvement to enhance the manuscript's quality. The reviewer recommends publication of this manuscript after the following comments have been addressed.

- What is the reason why Lanthanide metals have two times larger size than a carbon atom ? Then, other large elements can promote CO production in a similar way ?
- Please explain how the larger charge transfer leads to strong *COOH adsorption. Also explain that negligible charge transfer causes easier CO desorption.
- Based on ATR-IR results, it seems that RDS is the COOH-to-CO conversion. However, in DFT calculation result, COOH formation is shown to act as an RDS with the highest energy barrier. How can we understand the DFT and in situ results together ?
- In page 8 and line 4, the repetition of "CO₂ adsorption"
- In this manuscript, bridge adsorption on Er SAC is proposed to lower the energy barrier in the free energy diagram. But does Er SAC truly demonstrate a better onset potential for CO production compared to other catalysts, where key intermediates follow a single-line adsorption pathway? For instance, what if the authors compare Er SAC with Ni and Fe SACs?
- What is the main advantage of Ln SACs ? These require high overpotential, in spite of high FE within a wide potential window.

Reviewer #4

(Remarks to the Author)

The authors claim they developed a novel strategy utilizing atoms from the entire lanthanide group to facilitate the CO₂RR. The representative Er single atom catalyst exhibits a high Faradaic efficiency of CO exceeding 90%, a remarkable 42.2% full-cell energy efficiency and 70.4% single-pass CO₂ conversion efficiency. These results are significant. This paper is well organized, and the mechanism are successfully demonstrated by authors. This work can be published in Nature Communications. There are some minor problems or issues that should be considered as follows.

1. Why did the author design the Er SAC with a coordination number of 6 in the DFT section. As know, the coordination number is relative to the performance of SAC. So, how about other coordination numbers for Er SAC, such as 7. Also, the author chooses the main group Ca as the center of SAC. I noticed that the other main group Mg atom can be used in CO₂ reduction. The author should assess the Mg SAC performance in the DFT prediction.
2. The SAC with substrate of KB is synthesized in this work. The author should also provide the detail of preparation and suitable characterization results, such as TEM and SEM images.
3. Er SAC exhibits ~100 hours stability at 100 mA cm⁻². The authors should give some considerable characterization to analyze the state of catalyst after stability test and confirm the reason the performance decrease.
4. Other paper has reported the potential of different primary products (CH₄, CH₃OH) for Ln-based SACs (Langmuir 2023, 39, 41, 14748–14757). The author needs to check the NMR results about more than one SAC, such as Ce and Yb SAC.

Version 1:

Reviewer comments:

Reviewer #1

(Remarks to the Author)

The authors have improved the manuscript. The proposed bridge adsorbed *COOH configuration is still not that convincing,

in my opinion.

1. For my previous comment 2#, I was curious about the transition process from bridge adsorbed *COOH to linearly adsorbed *CO.

2. In the revised version, the authors report 70.4% single-pass CO₂ conversion efficiency at 200 mA cm⁻² with acidic electrolyte, which was obtained with 1 sccm CO₂ flow rate, as shown in Supplementary Table 6. Assuming a 100% CO Faradaic efficiency, the electroreduction of 0.7 sccm (1*70.4%) CO₂ to CO can only contribute a current of ~95 mA. Please double check this data. Can a high CO faradaic efficiency still be obtained at such a high CO₂ conversion efficiency with 1 sccm flow rate? Again, the full performance data set with Faradaic efficiencies at different flow rates should be provided. The specific flow rates for various cell measurements should be clearly provided in the experimental sections.

Reviewer #2

(Remarks to the Author)

The author has solved my doubts about the manuscript. It can be published without any changes.

Reviewer #3

(Remarks to the Author)

All the questions have been answered

Reviewer #4

(Remarks to the Author)

I am satisfied with the revised manuscript. Therefore, I am happy to recommend its acceptance as it is.

Version 2:

Reviewer comments:

Reviewer #1

(Remarks to the Author)

I find an important change of the electrode area in the equation (4), although the authors did not mention that in their rebuttal letter.

The electrode area used in the equation is 0.5 cm² in the current version of the manuscript, but it was 1 cm² in the last two versions! That's why I have large doubts on the high single-pass CO₂ conversion efficiency of 70.4%. If 0.5 cm² were correct, it would be mathematically fine.

In this regard, the values (100, 200) in Supplementary Table 6 should be current density with a unit of mA/cm², rather than current with a unit of mA. The expression "j (mA)" is also quite confusing, as j usually refers to current density.

The electrode area and/or effective electrode area and electrolyte used in MEA cell should also be provided clearly. Is that also 0.5 cm²?

The authors should be cautious and professional when presenting performance data.

Manuscript number: NCOMMS-24-64828-T

Title: Lanthanide Single-Atom Catalysts for Efficient CO₂-to-CO Electroreduction

A Point-to-Point Response to Reviewer's comments

Dear Editor and Reviewers,

Thank you for taking the time and effort to carefully examine our manuscript. The comments are highly appreciated and helpful to improve our work. We have made corresponding changes to the manuscript (highlighted in yellow) and supporting information to address the editor's concerns and the requests of the reviewers; these changes are specified and discussed in a point-to-point response to the editor's and reviewers' comments, as shown below:

REVIEWER COMMENTS

Reviewer #1 (Remarks to the Author):

This article reports a group of Ln MNC catalysts (up to 14 catalysts) for CO₂ electroreduction to CO, with all selective for CO production. While some of them have been previously, refs, 31,58-61, the authors claim that a key feature of these catalysts is the bridge adsorption of *COOH, which is different from common MNC with metals with low atomic number, such as Fe and Ca. If this were novel, the authors should validate their hypothesis with convincing experimental evidence, rather than only showing DFT results. Generally, this bridge adsorption is hardly stable, even from theoretical perspective. Furthermore, the presentation of the performance is very questionable (specific comments below). The CO₂ reduction performance is routine for MNC catalysts and indeed one could not see their advantages compared to common MNC catalysts such as Ni-N-C. Overall, the current manuscript is not suitable for publication in Nature Communications.

1. The bridge adsorption of *COOH, with both C and O bound with Ln metal atoms, is highlighted. My major concern, is such an adsorption mode really stable? It is rationalized by the authors by the large size of Ln atoms. But such a bridge adsorption

mode on larger metal atoms MNC such as Ir, Pt, Au have never been reported. If this is the case, why do not these MNC show similar performance? Convincing evidence on this key point should be provided.

Response: We highly appreciate the very constructive comments provided by the reviewer. To better answer this point, we have divided it into two parts.

1.A) is such an adsorption mode really stable?

The stable *COOH bridge adsorption on Ln SACs were also demonstrated in previous works (*Angew. Chem. Int. Ed.* **2020**, *59*,10651; *J. Mater. Chem. A*, **2024**, *12*, 1618). Moreover, *operando* ATR-IR spectra of Er SAC could detect the existence of *COOH bridge adsorption, consistent with the result of *COO bridge adsorption in the previous work (*J. Am. Chem. Soc.* **2023**, *145*, 11829). Furthermore, according to *operando* XANES spectra, the Er-N bond position of Er SAC shifted towards the Er-O side during CO₂RR, demonstrating the existence of Er-O from Er-*COOH bridge adsorption. Therefore, the *COOH bridge adsorption on Er site is stable in our design.

1.B) But such a bridge adsorption mode on larger metal atoms MNC such as Ir, Pt, Au have never been reported. If this is the case, why do not these MNC show similar performance?

Large size is important but not the only condition for *COOH bridge adsorption. We think there are three necessary conditions of Ln atoms to contribute to *COOH bridge adsorption, large size, oxophilicity and variable coordination numbers. The reasons as follows,

a) The large size of Ln atoms is important to mitigate steric-hindrance effects for double-line (bridge) adsorption on active sites.

b) The oxophilicity of Ln metals facilitates the formation of Metal-O bonding for *COOH, which is necessary to bridge adsorption. Based on previous studies (*Nature Commun.* **2023**, *14*, 3767; *Nat. Commun.* **2024**, *15*, 448), Ln elements exhibit strong oxophilicity which facilitates forming Ln-O for bridge adsorption of *COOH.

Moreover, the *operando* XANES spectra demonstrates Er-N bond position of Er SAC shifted towards the Er-O side during CO₂RR, demonstrating the existence of Er-O from Er-*COOH bridge adsorption.

c) The last but the most important condition is variable coordination numbers of Ln atoms. The transition from bridge adsorption of *COOH to linear adsorption of *CO means the coordination number of metal center changes during CO₂RR process. Hence, the variable coordination numbers of Ln atoms allow the formation of intermediates with different coordination numbers. Here is the reason for variable coordination numbers of Ln atoms: the coordinating stabilization energy (about 4.18 kJ mol⁻¹) of lanthanide atoms is much smaller than the crystal field stabilization energy of transition metals (typically ≥ 418 kJ mol⁻¹). Therefore, the coordinating bonds of lanthanide complexes are not directional, and the coordination number can vary from 3 to 12 (*Rare Earth Coordination Chemistry: Fundamentals and Applications*, John Wiley, **2010**, ISBN: 978-0-470-82485-6). This ensures the transition from bridge adsorption of *COOH to linear adsorption of *CO. Moreover, the coordination numbers can quickly change during electrocatalysis, facilitating the whole CO₂RR reaction (*Chem Catal.* **2022**, 2, 967).

As you suggest, the Ir, Pt and Au SACs were prepared using the same method as that of Ln SACs. As shown in **Figure R1**, the Ir SAC, Pt SAC and Au SAC all exhibit similar microstructures to Ln SACs. To evaluate the performance of Ir SAC, Pt SAC and Au SAC, the same electrochemical tests were conducted. As shown in **Figure R2**, Ir SAC, Pt SAC and Au SAC all show low activities to CO production, indicating large size of metal atoms is not the only condition to promote bridge adsorption of *COOH and thus CO production.

We also employed DFT calculations to further explore the deep reason. As exhibited in **Figure R3**, Pt and Au atoms don't have ability to form bridge adsorption of *COOH, mainly due to their highly stable coordination numbers, resulting in high energy barriers for *COOH formation. Besides, Ir SAC shows a larger barrier of *CO desorption. Hence, Ir SAC, Pt SAC and Au SAC all show poor performance of CO₂RR to CO.

To further explore whether bridge adsorption of *COOH on Ir SAC, Pt SAC and Au SAC is stable, we initially assumed *COOH bridge adsorption on Ir SAC, Pt SAC and Au SAC (**Figure R4**). After energy optimization, we found the bridge adsorption of *COOH turns into single line adsorption, demonstrating Ir SAC, Pt SAC and Au SAC are not capable of the *COOH bridge adsorption. On the contrary, *COOH bridge adsorption on Er SAC keep unchanged after optimization, demonstrating the stable *COOH bridge adsorption doesn't only depend on large size of metal atom.

To clearly present the strategy, we have added three conditions, large size, oxophilicity and variable coordination numbers, to our revised manuscript.

Figure R1. TEM, HRTEM and EDS mapping images for Ir SAC (**a**), Pt SAC (**b**) and Au SAC (**c**).

Figure R2. CO₂RR performance with 1 M KHCO₃ solution of Ir SAC (a), Pt SAC (b) and Au SAC (c). CO₂RR performance with 1 M KCl (pH=1) solution of Ir SAC (d), Pt SAC (e) and Au SAC (f)

Figure R3. Structure and adsorption configurations of key intermediates on Ir SAC (a), Pt SAC (b) and Au SAC (c). **d** Free energy diagram for CO₂ reduction to CO for metal SACs.

Figure R4. Initially assumed bridge adsorption configurations set of *COOH on Ir SAC (a), Pt SAC (b) and Au SAC (c). Final optimized stable *COOH adsorption configurations on Ir SAC (d), Pt SAC (e) and Au SAC (f)

2. *CO is still adsorbed linearly on Er-N-C as common metals. How does the transition from bridge adsorption to linear adsorption occur?

Response: Thanks for giving constructive comments on our studies. We think the ‘bridge-adsorption’ *CO can spontaneously turn into linear-adsorption *CO because linear-adsorption *CO is more stable. **Note:** the mentioned ‘bridge-adsorption’ *CO here shows as C and O atom bonding with Er atom, totally different from traditional *CO bridge adsorption (C atom bonding with two metal atom). The reason that the transition from bridge adsorption to linear adsorption as follows,

Generally, *CO adsorption on metal atom follows the “back-donation” mechanism (*J. Phys. Chem.* **68**, 2772; *J. Am. Chem. Soc.* **1985**, 107, 578), in which 5σ orbit of CO exhibits hybridization with metal atom first and then antibonding $2\pi^*$ orbits of CO show hybridization with occupied orbit of metal atom (**Figure R5**). Hence, the direction of C-O bond in *CO should be perpendicular to catalyst surface based on the direction of 5σ orbit, rather than as the pathway of ‘bridge-adsorption’.

To further verify this result, we initially assumed *CO ‘bridge adsorption’ on Er SAC (**Figure R6**). After energy optimization, we found the ‘bridge adsorption’ of *CO turns

into linear adsorption, demonstrating linear adsorption is more stable. Hence, the transition from bridge adsorption to linear adsorption on Er SAC occurs.

Figure R5. Schematic diagram of orbit hybridization between transition metal and CO.

Figure R6. a Structure and adsorption configurations of key intermediates on Er SAC. **b** Initially assumed bridge adsorption configurations set of *CO on Er SAC. **c** Final optimized stable *CO adsorption configurations on Er SAC.

3. The Er-N coordination number is 6. This is also unusual for MNC catalysts.

Response: Thanks for your helpful comments. To check synchrotron radiation fitting data, we have fitted the potential coordination structure with traditional coordination numbers, such as 4 and 5, to see if there are other possibilities. As shown in **Figure R7** and **Table R1**, the simulated Er-N₄ and Er-N₅ modes both show worse fitting goodness as compared to the experimental data due to the overlarge R-factors (The considerable R factor is smaller than 0.02). On the contrary, the fitting Er-N₆ modes models can accurately describe the actual experimental samples.

In fact, the coordinating bonds of lanthanide complexes are not directional, and the coordination number can vary from 3 to 12 (*Rare Earth Coordination Chemistry: Fundamentals and Applications*, John Wiley, **2010**, ISBN: 978-0-470-82485-6). We have summarized some Ln SACs in **Table R2**, and their coordination numbers are always 6 or more.

Figure R7. EXAFS fitting curves for Er SAC with different coordination numbers. **a** Er-N₄. **b** Er-N₅. **c** Er-N₆.

Table R1. First-shell fitting of FT EXAFS spectra with different coordination numbers for Er SAC.

	Shell	CN	R(Å)	σ^2	ΔE_0	R factor
Er-N ₄	Er-N	4.00	2.44	0.0038	4.85±3.42	0.052
Er-N ₅	Er-N	5.00	2.44	0.0014	4.47±2.37	0.029
Er-N ₆	Er-N	6.25	2.44	0.0038	4.00±1.89	0.018

CN: coordination numbers; R: bond distance; σ^2 : Debye-Waller factors; ΔE_0 : the inner potential correction. R factor: goodness of fit. S_0^2 was set to 0.87.

Table R2. Comparison of coordination numbers for Ln SACs.

Ln SACs	Coordination number in the first shell	Ref.
Er SAC	6	This work
La SAC	10	Sci. Adv. 2021, 7, 4915
La SAC	6	ACS Nano, 2020, 14, 15841
Er SAC	6	Angew.Chem. Int. Ed. 2020, 59, 10651
Sm SAC	6	Angew. Chem. Int. Ed. 2024, 63, e202405417
Ce SAC	10	ACS Catal. 2021, 11, 3923
Pr SAC	6	Chem. Eng. J. 2022, 436, 135271
Lu SAC	6	Small 2023, 2300926
Ce SAC	11	Adv. Mater. 2023, 35, 2302485

4. Why is the IR peak at 1800 to 1840 cm^{-1} assigned to bridge adsorption of *COOH? Can it be bridge adsorbed CO? The identification of the presence of the bridge-adsorbed *COOH intermediates is the most important point.

Response: Thanks for your helpful suggestion. *operando* ATR-IR spectra of Er SAC could directly detect the existence of *COOH bridge adsorption, consistent with the result of *COO bridge adsorption in the previous work (*J. Am. Chem. Soc.* **2023**, *145*, 11829). According to *operando* XANES spectra, the Er-N bond position of Er SAC

shifted towards the Er-O side during CO₂RR, demonstrating the existence of Er-O from Er-*COOH bridge adsorption. Therefore, the *COOH bridge adsorption on Er site is stable in our design.

We think the mentioned bridge-adsorbed *COOH here mainly include two pathways, ‘bridge-adsorption’—*CO show as C and O atom bonding with Er atom, and traditional *CO bridge adsorption— C atom bonding with two metal atoms. In **Question #2**, we have eliminated the possibility of ‘bridge-adsorption’ *CO because the ‘bridge-adsorption’ *CO can spontaneously turn into linear-adsorption *CO and linear-adsorption *CO is more stable. Hence, the IR peak in 1800 to 1840 cm⁻¹ can’t be assigned to ‘bridge adsorption’ of *CO.

We also checked the possibility of traditional *CO bridge adsorption. Firstly, the *operando* ATR-IR spectra of Er SAC was conducted with CO atmosphere. As shown in **Figure R8a**, no peak at 1800 to 1840 cm⁻¹ indicates *CO are difficult to adsorb on the Er SAC, not to mention traditional *CO bridge adsorption. Because Er element shows as atomically dispersed state, traditional *CO bridge adsorption on single atom cannot be achievable.

In addition, we take the only possibility, Er metal, into consideration. Using the Density Functional Perturbation Theory (DFPT) calculations with VASP (**Figure R8b**), the predicted position of *CO bridge adsorption on Er(001) was 1706 cm⁻¹, which is much lower than 1800 cm⁻¹, further excluding the possibility of *CO bridge adsorption. Hence, the IR peaks at 1800 to 1840 cm⁻¹ can’t be assigned to bridge adsorption of *CO.

Figure R8. a. *Operando* ATR-IR spectra of Er SAC with CO atmosphere. **b.** The calculated models and their corresponding vibrational wavenumber for bridge *CO adsorption on Er(001) by DFPT calculations.

For CO adsorption on the Er(001) surface, a 3×3 supercell consisting of four layers was constructed. During the structural optimization, the bottom two layers were fixed to mimic the bulk properties. The IR spectrum was then calculated using VASP, where the slab atoms were kept fixed, and the frequencies of the adsorbates were computed using DFPT. The script from <https://github.com/dakarhanek/VASP-infrared-intensities> was utilized to extract the final IR peaks and intensities.

5. Performance presentation

1). The statement “Such $\geq 90\%$ Faradaic efficiency at 500 mA/cm^2 both in neutral (top panel, Fig. 4b) and acidic electrolyte (bottom panel, Fig. 4b and Supplementary Fig. 43) endows the SAC a high turnover frequency (TOF) of $\sim 130,000 \text{ h}^{-1}$, simultaneously

representing an unprecedented full-cell energy efficiency of 42.2% (Supplementary Fig. 44-46 and table 4)” is quite confusing. At which current density, potential and electrolyte the TOF was calculated? And, more important, the so-called high full cell energy efficiency of 42.2% is actually obtained at 50 mA/cm² (Supplementary Fig. 44). All the performance metrics should be presented in a clear way. In Supplementary Table 4, the corresponding current densities should be clearly provided. On Page 15, full-cell voltage (V_{full}) was used in the calculation of energy efficiency. How was the full-cell voltage measured in a flow cell with three-electrode system?

Response: Thanks for your helpful suggestion. The turnover frequency (TOF) of h⁻¹ 129,204 h⁻¹ and ~130,783 h⁻¹ could be obtained at 500 mA/cm² in neutral and acidic electrolyte, respectively. After approximate treatment, we made the statement ‘Such \geq 90% Faradaic efficiency at 500 mA/cm² both in neutral (top panel, Fig. 4b) and acidic electrolyte (bottom panel, Fig. 4b and Supplementary Fig. 43) endows the SAC a high turnover frequency (TOF) of ~130,000 h⁻¹.’ Following your suggestion, we have clearly added more details for the statement and in the revised manuscript and supplementary Tables (**Figure R3**):

Such \geq 90% Faradaic efficiency at 500 mA cm⁻² both in neutral (top panel, Fig. 4b) and acidic electrolyte (bottom panel, Fig. 4b and Supplementary Fig. 43) endows the SAC a high turnover frequency (TOF) of ~130,000 h⁻¹. Furthermore, Er SAC can achieve an unprecedented full-cell energy efficiency of 34.7% at 200 mA cm⁻² (Supplementary Fig. 44-47 and Table 4).⁵³⁻⁵⁶

We employed an external voltmeter to measure the full-cell voltage of a flow cell with a three-electrode system. As shown in **Figure R9**, the voltmeter can directly measure the full-cell voltage of a flow cell, because the current through the reference electrode is close to zero. We have added the details to our revised manuscript. Moreover, to decrease resistance of the system, we also have tested the performance in the Membrane Electrode Assembly (MEA). As shown in **Table R4**, MEA fabricated Er SAC shows high full-cell energy efficiency of ~34.7% even at current density of 200 mA cm⁻². All revised details have been provided in the revised manuscript.

Table R3. The full-cell energy efficiency (EE) of CO₂ to CO for Er SAC in flow-cell with acidic and neutral electrolyte.

Acidic electrolyte (Flow cell)				Neutral electrolyte (Flow cell)			
Current density (mA cm ⁻²)	Voltage (V)	FE _{CO} (%)	EE (%)	Current density (mA cm ⁻²)	Voltage (V)	FE _{CO} (%)	EE (%)
50	3.15	99.2	42.2	50	4.21	99.8	31.7
100	3.99	99.2	33.3	100	6.05	94.6	21.0
200	5.01	99.7	26.7	200	9.72	93.9	13.0
300	6.04	98.2	21.8	300	12.71	93.2	9.8
400	6.60	96.4	19.6	400	15.51	90.0	7.8
500	8.21	91.1	14.9	500	16.70	90.0	7.2

Figure R9. Schematic diagram for the full-cell voltage measurement in a flow cell with three electrode system.

Table R4. The full-cell energy efficiency (EE) of CO₂ to CO for Er SAC in MEA with acidic electrolyte.

Acidic electrolyte (MEA)			
Current density (mA cm ⁻²)	Voltage (V)	FE _{CO} (%)	EE (%)
50	3.08	99.3	43.2
100	3.43	99.1	38.7
200	3.76	97.3	34.7
300	3.98	96.4	32.5
400	4.17	93.1	29.9
500	4.42	90.2	27.3

2). It is important to indicate at which current density the energy efficiency and carbon efficiency were obtained, in Supplementary Table 6.

Response: Thanks for your suggestion. Following your suggestion, we have provided current density for all the energy efficiency and carbon efficiency in the revised manuscript and supporting information. Because the best performance about energy efficiency and carbon efficiency in many works are not with same condition, we have divided the comparison into two tables (Table R5 and R6).

Table R5. Comparison of full-cell energy efficiency of various catalysts during CO₂ electroreduction.

Catalysts	Current density (mA/cm ⁻²)	Full-cell energy efficiency (%)	Ref.
Er SAC-Flow cell	50	42.2	This work
Er SAC-MEA	50	43.2	This work
Er SAC-MEA	100	38.7	This work
Er SAC-MEA	200	34.7	This work
CG-medium Cu	100	28	Nat. Catal. 2023, 6, 763
PFSA modified Cu	200	25	Nat. Synth. 2023, 2, 403

NiNC-IMI	200	40	Nat. Chem. Eng. 2024, 1, 229
EC-Cu	200	30	Nat. Commun. 2023, 14, 2387
Co-CNTs-MW	200	54.1	Nat. Commun. 2023, 14, 1599
Cu _{0.9} Zn _{0.1}	150	31.2	Nat. Commun. 2023, 14, 1298
F-CuNS	800	10	Nat. Commun. 2022, 13, 7596

Table R6. Comparison of single-pass carbon efficiency of various catalysts during CO₂ electroreduction in acidic electrolyte.

Catalysts	Current density (mA/cm ²)	Single-pass carbon efficiency (%)	Ref.
Er SAC	200	70.4	This work
CG-medium Cu	100	90	Nat. Catal. 2023, 6, 763
Pd-Cu	500	62	Nat. Catal. 2022, 5, 564
PFSA modified Cu	200	75	Nat. Synth. 2023, 2, 403
CoTAAPc@CNT-n	256	67.4	Nat. Synth. 2024, 3, 1231
NiNC-IMI	200	40	Nat. Chem. Eng. 2024, 1, 229
Cu-GDL	500	42	Nat. Commun. 2024, 15, 491
EC-Cu	200	70	Nat. Commun. 2023, 14, 2387
Sn(S)-H	400	36	Nat. Commun. 2023, 14, 2843
Co-CNTs-MW	100	40.4	Nat. Commun. 2023, 14, 1599
Cu _{0.9} Zn _{0.1}	400	31	Nat. Commun. 2023, 14, 1298
F-CuNS	800	54	Nat. Commun. 2022, 13, 7596

3). It is also important to indicate with which kind of cell the performance was evaluated, in Supplementary Table 7, as the H-cell performance is for sure much lower than flow cell performance for a given catalyst. Obtaining an ultrahigh TOF is routine, using gas diffusion electrode with a low catalyst loading, especially for an element with high atom number.

Response: Thanks for your comment. Following your suggestion, we have indicated which kind of cell and evaluated the TOF both in H-cell and flow cell. As shown in **Table R7**, Er SAC also achieves an ultrahigh TOF of ~60,000 h⁻¹ at with H-cell -0.97

V vs. RHE, compared to other SACs. Hence, Er SAC shows remarkable TOFs both in H-cell and flow cell.

Table R7. Comparison of CO₂-to-CO TOF of the state-of-art catalysts in H-cell and flow cell.

Catalysts	The Maximal TOF (hour⁻¹) under FE_{CO} ≥ 90%	Cell type	Ref.
Er SAC	130,000	Flow cell	This work
NiPc-OMe MDE	43,200	Flow cell	Nat. Energy 2020, 5, 684
Ni ₂ NC	77,500	Flow cell	Nat. Synth. 2022, 1, 719
CoCu-DASC	91,458	Flow cell	Angew. Chem. Int. Ed. 2022, 61, e202212329
Er SAC	60,000	H-cell	This work
Fe ³⁺ -N-C	1,000	H-cell	Science 2019, 364, 1091
A-Ni-NSG	14,800	H-cell	Nat. Energy 2018, 3, 140
Co-S ₁ N ₃	4,564	H-cell	Nat. Commun. 2024, 15, 416
Fe-poN-C/Fe	2,890	H-cell	Nat. Commun. 2023, 14, 5108
Co-CNTs-MW	25,896	H-cell	Nat. Commun. 2023, 14, 1599
TeN ₂ -CuN ₃	24,080	H-cell	Nat. Commun. 2023, 14, 6164
Ni@C ₃ N ₄ -CN	22,000	H-cell	Nat. Commun. 2022, 13, 6082
NiFe-DASC	15,055	H-cell	Nat. Commun. 2021, 12, 4088
Ni-N ₂ C ₂	3,903	H-cell	Nat. Commun. 2020, 11, 2256
(Cl, N)-Mn/G	38,347	H-cell	Nat. Commun. 2019, 10, 2980
Ni-NCB	36,000	H-cell	Joule 2019, 3, 265,

4). NiNC is generally considered the most active among MNC catalysts with common metal elements. Is the ErNC better than NiNC? Please show such a performance comparison in a fair way, also with current density and FE.

Response: Thanks for your comment. Following your suggestion, we have prepared the Ni SAC using the same method as Ln SACs. As shown in **Figure R10**, all exhibit similar microstructures to Ln SACs. To evaluate the performance of Ni SAC, the same electrochemical evaluation was conducted. As exhibited in **Figure R11**, Ni SAC shows a lower efficiency (<70%) of CO at 500 mA cm⁻² both in acidic and neutral electrolyte than those of Er SAC ($\geq 90\%$). It is worth noting that both have ultralow metal atomic content (~ 0.2 at%), but Er SAC can achieve higher FE_{CO} (>90%) than Ni SAC at high current densities, highlighting the advantage of Ln SAC.

Figure R10. a TEM, b HRTEM and c EDS mapping images for Ni SAC.

Figure R11. CO₂RR performance of Ni SAC with 1 M KHCO₃ solution (a) and 1 M KCl (pH=1) solution (b).

Reviewer #2 (Remarks to the Author):

The manuscript "Lanthanide Single-Atom Catalysts for Efficient CO₂-to-CO Electroreduction" by Q. Wang et al provides a comprehensive study of the synthesis and evaluation of lanthanide group single atom catalysts for CO₂RR, especially elucidating the bridging adsorption mechanism. Also, the authors applied a vast variety of methods to study the materials including theoretical calculation, physical characterization as well as electrochemical analysis. I recommend that this work can be published in Nature Communications after addressing the following minor issues:

1. Fe SAC shows a low CO selectivity in electrochemical CO₂ reduction especially at -0.57 V. Moreover, since the bubbles from HER reaction would weaken the signals of *CO, it should be difficult to observe the peak of *CO. However, an obvious peak ascribed to the adsorption of linear *CO on the Fe site could be observed in the operando ATR-IR spectra.

Response: Thank you for this comment. We agree with your opinion that these bubbles would affect the quality of the ATR-IR data. Hence, to collect clearer images about the absorption of linear *CO, we used *operando* attenuated total reflection surface-enhanced infrared absorption spectroscopy (ATR-IR) to amplify the absorbance of CO molecules vibrations (*ACS Catal.* **2021**, 11, 840–848; *ACS Energy Lett.* **2019**, 4, 682–689). As shown in **Figure R12**, we prepared ultrathin Au films in Si electrodes to produce surface plasmon for strong signals. This mode can be used to observe traces of *CO.

Furthermore, ATR-IR is a short-range test method, which can suppress the interference of H₂ gas. The depth of penetration, d_p , can be calculated with a simple equation according to Harrick:

$$d_p = \frac{\lambda}{2\pi n_1 (\sin^2 \theta - \left(\frac{n_2}{n_1}\right)^2)^{1/2}}$$

where λ is the wavelength of the incident radiation, θ is the angle of incidence of the

incident radiation, n_1 is the refractive index of the IRE and n_2 is the refractive index of the rare medium (*Progress in Organic Coatings* **2006**, 57, 78–88). For Au films and a measurement angle of 65° , the depth of penetration in the spectral region between 1600 and 2400 cm^{-1} would only limit in a few microns (μm) from the electrode surface. Because H_2 is quickly desorbed from the electrode surface to the bulk of electrolyte, the interference of gas is greatly reduced.

Using ATR-SEIRAS, Gong et al. and Shao et al. all observed the signal of $^*\text{CO}$, even though their catalysts showed very low CO faradaic efficiency at certain potentials (*Angew. Chem. Int. Ed.* **2021**, 133, 15472–15475; *ACS Energy Lett.* **2019**, 4, 1778–1783).

Figure R12. Schematic representation of *operando* attenuated total reflection surface-enhanced infrared absorption spectroscopy.

2. The pure CNTs should be analyzed in detail. Since the contents of single metal atom is very low and it indeed influences the performance of CO_2RR , the impurities of the raw materials should be eliminated thoroughly. Some detection such as thermogravimetric analysis should be operated to support the conclusion.

Response: Thanks for your constructive suggestion. Before using the CNT, we used 0.5 M HNO_3 to remove the potential metal impurities at 80°C for 12 hours. Furthermore,

to exclude all the potential sources of contamination, we first measured the metal content of CNT through Inductively Coupled Plasma Mass Spectrometry (ICP-MS, Agilent 7700s) directly. As is shown in **Table R8**, CNT does not contain metal impurities, including Fe, Co, Ni, Cu, Zn and Mn elements.

Table R8. The contents of metal in CNT.

Element	Mn	Fe	Co	Ni	Cu	Zn
Content (%)	0	0	0	0	0	0

The content below 0.001% is negligible.

Moreover, to further obtain the image of potential metal contamination, we have also employed high-resolution XPS. As is shown in **Figure R13**, no XPS signal of Mn 2*p*, Fe 2*p*, Ni 2*p*, Co 2*p*, Cu 2*p*, Zn 2*p* and In 2*p* is detected in CNT.

Figure R13. High-resolution XPS spectra of Mn 2*p*, Fe 2*p*, Co 2*p*, Ni 2*p*, Cu 2*p*, and Zn 2*p* for the raw materials.

Furthermore, we also analyzed the CO₂RR performance of pure CNT (**Figure R14**). The pure CNT exhibits almost 100% H₂ selectivity at all potentials, manifesting CNT only works as a substrate and does not have a decisive effect on the performance of

SAC.

Figure R14. Characterizations and CO₂RR test of CNT. **a** XRD patterns. **b** Raman spectra. **c** SEM image. **d** HRTEM image. **e** EDS mappings for pure CNT. **f** LSV curves of pure CNT. **g** FE_{H₂} of pure CNT at different potentials in H-cell.

As you suggested, we have conducted thermogravimetric analysis (TGA) to explore the catalyst (**Figure R15**). These results are supplemented in the revised manuscript and supporting information.

Figure R15. TGA curves of different catalysts in air.

3. The FT-IR spectra of Er SAC should be provided to exclude the existence of C_3N_4 and the Er_2O_3 PDF of XRD in supporting information to prove the results regarding the feature of atomic dispersion.

Response: Thanks for your constructive suggestion. We have conducted the FT-IR spectra for Er SAC. As shown in **Figure R16**, the characteristic peak of C_3N_4 disappeared after pyrolysis.

Figure R16. FT-IR spectra of C_3N_4 NS and Er SAC.

We have also added the Er_2O_3 PDF of XRD to supporting information. As exhibited in **Figure R17**, no Er_2O_3 is found in these metal SACs. These results are supplemented in the revised manuscript and supporting information.

Figure R17. XRD patterns of catalysts.

4. In this work, Er SAC achieves durability at -100 mA/cm^2 for 100 hours. Why did this performance of flow cell have drop after 100 h. Is there a destruction in the Er single atom catalyst, such as dissolution or aggregation? Or did the catalyst break off and flooding occur during the flow-cell test? The author should find the reason carefully.

Response: Thank you for the comment. We have carefully checked and corrected the errors. The declining performance of flow cell is mainly due to the infiltration of gas diffusion layer (GDE) by electrolyte flood. As known, the stability of catalyst is highly relative to the hydrophobicity of gas diffusion layer. The CO₂ diffusion in flow cell will be intercepted by the electrolyte flood due to its decreased hydrophobicity during a successive electrocatalysis (**Figure R18a**). We have conducted the XRD patterns, dark-field transmission electron microscopy (TEM) and EDS mapping for the Er SAC after stability test. As shown in **Figure R18**, no aggregation is found, and Er single atoms still exist in the CNT substrate.

Figure R18. a Gas diffusion electrode flooding after long-term operation in flow cell.

b XRD pattern of Er SAC after stability test. **c** Dark-field TEM and EDS mapping image of Er SAC after stability test.

5. Some minor errors should be corrected carefully, such as the format in formula 4, 5, The titles of horizontal axis in Figure 2f and 3d, and the useless line in Figure S3.

Response: Thank you for these suggestions. We have carefully checked and corrected the errors.

Reviewer #3 (Remarks to the Author):

This manuscript reports the universal properties of Lanthanide SACs for CO production via the electrochemical reduction of carbon dioxide. The underlying principle is intriguing and is expected to attract considerable interest from readers in the field of catalysis. However, several issues still need improvement to enhance the manuscript's quality. The reviewer recommends publication of this manuscript after the following comments have been addressed.

1. What is the reason why Lanthanide metals are two times larger size than a carbon atom? Then, other large elements can promote CO production in a similar way?

Response: Thank you for these comments. To better answer, we have divided them into two parts.

1) What is the reason why Lanthanide metals are two times larger size than a carbon atom?

As we know, the more electron layers outside nucleus, the larger size (radius) of atom. The result that lanthanide metals are two times larger size than a carbon atom refers to the scientists, J. C. Slater (*J. Chem. Phys.* **1964**, *41*, 3199) and Enrico Clementi (*J. Chem. Phys.* **1967**, *47*, 1300). The detail as follows,

Generally, there are four widely used definitions of atomic radius: Van der Waals radius, ionic radius, metallic radius and covalent radius. Based on our characterization, the Ln and carbon atoms in Ln SACs bond with other atom as the pathway of covalence. According to empirically measured covalent radius for the elements by J. C. Slater (*J. Chem. Phys.* **1964**, *41*, 3199), the radiuses of Ln atoms are 175-195 pm (picometers), two times larger than that of carbon atom (70 pm). Moreover, Enrico Clementi et al. presented the atomic radius computed from theoretical models in 1967 (*J. Chem. Phys.* **1967**, *47*, 1300). The radiuses of Ln atoms are 205-238 pm (picometers), also two times larger than that of carbon atom (67 pm).

2) Other large elements can promote CO production in a similar way?

Large size is important but not the only condition for *COOH bridge adsorption. We think there are three necessary conditions of Ln atoms to contribute to *COOH bridge adsorption, large size, oxophilicity and variable coordination numbers. The reasons as follows,

a) The large size of Ln atoms is important to mitigate steric-hindrance effects for double-line (bridge) adsorption on active sites.

b) The oxophilicity of Ln metals facilitates the formation of Metal-O bonding for *COOH, which is necessary to bridge adsorption. Based on previous studies (*Nature Commun.* **2023**, *14*, 3767; *Nat. Commun.* **2024**, *15*, 448), Ln elements exhibit strong oxophilicity which facilitates forming Ln-O for bridge adsorption of *COOH.

c) The last but the most important condition is variable coordination numbers of Ln atoms. The transition from bridge adsorption of *COOH to linear adsorption of *CO means the coordination number of metal center changes during CO₂RR process. Hence, the variable coordination numbers of Ln atoms allow the formation of intermediates with different coordination numbers. Here is the reason for variable coordination numbers of Ln atoms: the coordinating stabilization energy (about 4.18 kJ mol⁻¹) of lanthanide atoms is much smaller than the crystal field stabilization energy of transition metals (typically ≥ 418 kJ mol⁻¹). Therefore, the coordinating bonds of lanthanide complexes are not directional, and the coordination number can vary from 3 to 12 (*Rare Earth Coordination Chemistry: Fundamentals and Applications*, John Wiley, **2010**, ISBN: 978-0-470-82485-6). This ensures the transition from bridge adsorption of *COOH to linear adsorption of *CO. Moreover, the coordination numbers can quickly change during electrocatalysis, facilitating the whole CO₂RR reaction (*Chem Catal.* **2022**, *2*, 967).

For well comparison, we select other large elements to further verify the results, such as Ir (180 pm), Pt (177 pm) and Au (174 pm). The Ir SAC, Pt SAC and Au SAC were prepared using the same method as that of Ln SACs. As shown in **Figure R19**, the Ir SAC, Pt SAC and Au SAC all exhibit similar microstructure to Ln SACs.

To evaluate the performance of Ir SAC, Pt SAC and Au SAC, the same electrochemical tests were conducted. As shown in **Figure R20**, Ir SAC, Pt SAC and Au SAC all show ultralow activity to CO production, indicating large size of metal atoms is not only condition to promote bridge adsorption of *COOH and thus CO production. We also employed DFT calculations to further explore the deep reason. As exhibited in **Figure R21**, Pt and Au atoms don't show ability to form bridge adsorption of *COOH, mainly due to their highly stable coordination numbers, resulting in high energy barrier for *COOH formation. Besides, Ir SAC shows a larger barrier of *CO desorption. Hence, Ir SAC, Pt SAC and Au SAC all show poor performance of CO₂RR to CO.

To more clearly present the strategy, we have added three conditions, large size, oxophilicity and variable coordination numbers, to our revised manuscript.

Figure R19. TEM, HRTEM and EDS mapping images for Ir SAC (a), Pt SAC (b) and Au SAC (c).

Figure R20. CO₂RR performance with 1 M KHCO₃ solution of Ir SAC (a), Pt SAC (b) and Au SAC (c). CO₂RR performance with 1 M KCl (pH=1) solution of Ir SAC (d), Pt SAC (e) and Au SAC (f)

Figure R21. Structure and adsorption configurations of key intermediates on Ir SAC (a), Pt SAC (b) and Au SAC (c). **d** Free energy diagram for CO₂ reduction to CO for metal SACs.

2. Please explain how the larger charge transfer leads to strong *COOH adsorption. Also explain that negligible charge transfer causes easier CO desorption.

Response: Thanks for your comments. The nature of *COOH and *CO adsorption on metal sites is the formation of covalent bonds, which results in electron or charge transfer between intermediates and metal sites (*Adv. Mater.* **2023**, *35*, 2300695; *Angew. Chem. Int. Ed.* **2021**, *60*, 25241). Generally, for the same intermediate, larger charge transfer between intermediates and metal sites indicates stronger chemical bond and intermediates adsorption.

Hence, based on Charge density differences and Bader charge analysis, charge transfer from Er and Fe SAC to *COOH is 0.500 and 0.584 *e* respectively, larger than that of Ca SAC (0.363 *e*), indicating strong *COOH adsorption on Er sites by virtue of bridge adsorption. Furthermore, Er SAC (0.012 *e*) and Ca SAC (0.005 *e*) provide negligible charge to *CO, compared to Fe SAC (0.136 *e*), demonstrating the easier CO desorption on Er and Ca site than that on Fe site.

3. Based on ATR-IR results, it seems that RDS is the COOH-to-CO conversion. However, in DFT calculation result, *COOH formation is shown to act as an RDS with the highest energy barrier. How can we understand the DFT and in situ results together?

Response: Thanks for the constructive comments. Since the formation and consumption of *COOH occur simultaneously, the difference in the amount of *COOH adsorption at different times could be obtained directly through comparing the rates of formation and consumption. If *COOH-to-*CO conversion is shown to act as an RDS with the highest energy barrier, the amount of *COOH adsorption should be increased as the time goes by. Based on this point, we have conducted time-varying ATR-IR for Er SAC at same potential. As shown in **Figure R22**, the intensity of *COOH remains unchanged after 5 min, indicating the amount of *COOH doesn't increase on Er sites. This result demonstrates *COOH to *CO is not hindered by the highest energy barrier. In fact, the hydrogenation of *COOH to *CO is considered as spontaneous reactions,

much easier than the hydrogenation of CO₂. (*ACS Catal.* **2020**, *10*, 10068; *Nat. Commun.* **2017**, *8*, 944).

The *operando* ATR-IR spectra aim to amplify the absorbance of intermediate vibration, which is sensitive to strong adsorbed intermediates (*ACS Catal.* **2021**, *11*, 840–848; *ACS Energy Lett.* **2019**, *4*, 682–689). Hence, the remarkable peaks manifest the bridge adsorption of *COOH on Er SAC is strong to detect. For instance, the peak of *COOH and *CO on Fe SAC are both remarkable to detect, proving the strong *COOH and *CO on Fe sites. Hence, *in situ* results demonstrate Er SAC show facilitated a *COOH adsorption ability, well consistent with that from that from DFT calculations.

Figure R22. *Operando* ATR-IR spectra of Er SAC at -0.67 V before and after 5 min.

4. In page 8 and line 4, the repetition of "CO₂ adsorption"

Response: Thanks for your suggestion. We have revised this error carefully.

5. In this manuscript, bridge adsorption on Er SAC is proposed to lower the energy barrier in the free energy diagram. But does Er SAC truly demonstrate a better onset potential for CO production compared to other catalysts, where key intermediates follow a single-line adsorption pathway? For instance, what if the authors compare Er SAC with Ni and Fe SACs?

Response: Thanks for your helpful comment. Following your suggestion, we have calculated the onset potential of Er and Fe SACs. As shown in **Figure R23**, the onset potential ($j_{co} = 0.1 \text{ mA cm}^{-2}$) of Er SAC is $\sim -0.31 \text{ V vs. RHE}$ more positive than that of Fe SAC ($\sim -0.36 \text{ V vs. RHE}$) (*Curr. Opin. Electrochem.* **2023**, *37*, 101176), proving the bridge adsorption indeed lowers the energy barrier of *COOH formation.

Figure R23. CO partial current density over different catalysts in pure CO₂ saturated 0.5 M KHCO₃ solution in H-cell.

But in the original work, we also think Fe SAC also possesses a prominent ability to *COOH formation even though it shows a single-line adsorption pathway for *COOH, because the *d*-block SACs have directionally localized *d* orbitals to strongly interact with or adsorb *COOH (*Angew. Chem. Int. Ed.* **2021**, *60*, 25241; *Adv. Mater.* **2023**, *35*, 2300695). Hence, the calculated energy barriers for *COOH formation (0.61 eV of Er SAC and 0.63 eV of Fe SAC), and the onset potentials for CO production on Er and Fe sites are very close.

Noticeably, compared to Fe SAC, Er SAC shows a much better performance for *CO desorption. The energy barrier of *CO desorption from Er sites (0.34 eV) is much lower than that from Fe sites (1.06 eV), indicating *CO would easily poison the Fe sites. Moreover, the *CO desorption from active sites is a thermodynamic process without

electron transfer behavior, suggesting the change of potential has no effect on *CO desorption. As a result, the *d*-block SACs often show poor performances at large current densities due to the large amounts of CO generation poisoning the metal sites (*Angew. Chem. Int. Ed.* **2021**, *60*, 25241).

In our work, we have demonstrated Fe SAC shows a poorer performance of CO production at large current densities whether in acidic or neutral electrolyte. To further demonstrate this result, we have prepared the Ni SAC using the same method as Ln SACs. As shown in **Figure R24**, all exhibit similar microstructures to Ln SACs. To evaluate the performance of Ni SAC, the same electrochemical evaluation was conducted. As exhibited in **Figure R25**, Ni SAC shows a lower efficiency (<70%) of CO at 500 mA cm⁻² than that Er SAC ($\geq 90\%$), further demonstrating our conclusion. This advantage in large current densities endows Er SAC could reach an ultrahigh TOF (turnover frequency), even compared to other state-of-art Ni single-atom catalysts (**Table R9**).

Figure R24. a TEM, b HRTEM and c EDS mapping images for Ni SAC.

Figure R25. CO₂RR performance Ni SAC with 1 M KHCO₃ (a) and 1 M KCl (pH=1) solution (b).

Table R9. Comparison of CO₂-to-CO TOF of the state-of-art Ni single-atom catalysts.

Catalysts	The Maximal TOF (hour ⁻¹) under FE _{CO} ≥ 90%	Ref.
Er SAC	130,000	This work
A-Ni-NSG	14,800	Nat. Energy 2018 , 3, 140
NiPc-OMe MDE	43,200	Nat. Energy 2020 , 5, 684
Ni ₂ NC	77,500	Nat. Synth. 2022 , 1, 719
Ni@C ₃ N ₄ -CN	22,000	Nat. Commun. 2022 , 13, 6082
Ni-NCB	36,000	Joule 2019 , 3, 265,
Ni/Cu-N-C	20,695	J. Am. Chem. Soc. 2022 , 144, 9661
NiFe-DASC	15,055	Nat. Commun. 2021 , 12, 4088
ZIF-NC-Ni-Fe	2,210	Angew. Chem. Int. Ed. 2022 , e202205632
Ni-N ₃ -C	1,425	Angew. Chem. Int. Ed. 2021 , 60, 7607
Ni SAs/OMMNC	11,000	Energy Environ. Sci. , 2023 , 16, 502
Ni@CC-T	22,489	Adv. Mater. 2023 , 35, 2205553

6. What is the main advantage of Ln SACs? These require high overpotential, in spite of high FE within a wide potential window.

Response: Thanks for your constructive comment and we fully understand this concern about the overpotential of Ln SACs. We are sorry to overlook this issue in our paper, because the total current for CO₂ reduction at more positive potentials is too low to accurately calculate the FE_{CO} (GC limitation). As shown in **Figure R26**, Er SAC can reach a FE_{CO} of 90% at -0.37 V vs. RHE, verifying Er SAC has a lower overpotential than that in the original version. However, Fe and Ca SACs show very low total current (< 0.2 mA cm⁻² with electrode area of 0.25 cm²) at -0.37 V vs. RHE. Hence, to better compare Ln SACs with Fe and Ca SACs, we only selected the considerable potential window to test the performance.

Figure R26. FE_{CO} of Er SAC at different potentials in H-cell under pure CO₂.

Taking Er SAC as an example, we think the main advantages of Ln SACs mainly include high TOF (turnover frequency), robust stability, high energy efficiency and high SPCE at high current density

1) **high TOF.** Er SAC needn't suffer CO poison issue. Even with low metal atom content (0.2 at%), Er SAC could reach high FE_{CO} (>90%) at high current densities. This large atom utilization endows Er SAC possessing an ultrahigh TOF of ~130,000 h⁻¹ for CO₂RR to CO, compared to the state-of-art catalysts (**Table R10**), introducing potential benefits for practical applications.

Table R10. Comparison of CO₂-to-CO TOF of the state-of-art catalysts in H-cell and flow cell.

Catalysts	The Maximal TOF (hour ⁻¹) under FE _{CO} ≥ 90%	Cell type	Ref.
Er SAC	130,000	Flow cell	This work
NiPc-OMe MDE	43,200	Flow cell	Nat. Energy 2020, 5, 684
Ni ₂ NC	77,500	Flow cell	Nat. Synth. 2022, 1, 719
CoCu-DASC	91,458	Flow cell	Angew. Chem. Int. Ed. 2022, 61, e202212329
Er SAC	60,000	H-cell	This work
Fe ³⁺ -N-C	1,000	H-cell	Science 2019, 364, 1091
A-Ni-NSG	14,800	H-cell	Nat. Energy 2018, 3, 140
Co-S ₁ N ₃	4,564	H-cell	Nat. Commun. 2024, 15, 416
Fe-poN-C/Fe	2,890	H-cell	Nat. Commun. 2023, 14, 5108
Co-CNTs-MW	25,896	H-cell	Nat. Commun. 2023, 14, 1599
TeN ₂ -CuN ₃	24,080	H-cell	Nat. Commun. 2023, 14, 6164
Ni@C ₃ N ₄ -CN	22,000	H-cell	Nat. Commun. 2022, 13, 6082
NiFe-DASC	15,055	H-cell	Nat. Commun. 2021, 12, 4088
Ni-N ₂ C ₂	3,903	H-cell	Nat. Commun. 2020, 11, 2256
(Cl, N)-Mn/G	38,347	H-cell	Nat. Commun. 2019, 10, 2980
Ni-NCB	36,000	H-cell	Joule 2019, 3, 265,

2) **high single-pass CO₂ conversion efficiency.** The remarkable performance in CO₂ activation or *COOH formation endows Er SAC could reach high FE_{CO} >90% even at low CO₂ flow rates (1 sccm), ensuring high single-pass CO₂ conversion efficiency (70.4%) (Figure R27, Table R9).

Figure R27. FE_{CO} and current densities for Er SAC at different CO₂ flow rates at 1 M KCl (pH=1).

Table R9. Comparison of full-cell energy efficiency and single-pass carbon efficiency of various catalysts during CO₂ electroreduction.

Catalysts	Full-cell energy efficiency (%)	Single-pass carbon efficiency (%)	Ref.
Er SAC	42.2	70.4	This work
PFSA-modified Cu	8	77	Science 2021, 372, 1074
CG-medium Cu	28	90	Nat. Catal. 2023, 6, 763
Pd-Cu	-	62	Nat. Catal. 2022, 5, 564
PFSA modified Cu	25	75	Nat. Synth. 2023, 2, 403
NiNC-IMI	40	40	Nat. Chem. Eng. 2024, 1, 229
Cu-GDL	-	42	Nat. Commun. 2024, 15, 491
EC-Cu	20	70	Nat. Commun. 2023, 14, 2387
Sn(S)-H	-	36	Nat. Commun. 2023, 14, 2843
Co-CNTs-MW	54.1	40	Nat. Commun. 2023, 14, 1599
Cu _{0.9} Zn _{0.1}	32	33	Nat. Commun. 2023, 14, 1298
F-CuNS	-	54	Nat. Commun. 2022, 13, 7596

3) **robust stability.** The flow cell fabricated with Er SAC shows a maintained stable

operation at 100 mA cm^{-2} , with $\text{FE}_{\text{CO}} > 90\%$ in acidic electrolyte (100 h). No obvious cluster and Er-Er bond are observed on Er SAC after the test according to *operando* XANES spectra and TEM images, demonstrating the durability of structure.

4) **high energy efficiency.** The flow cell fabricated with Er SAC shows an unprecedented full-cell energy efficiency of 42.2%. Moreover, to decrease resistance of the system, we have tested the performance in the Membrane Electrode Assembly (MEA). As shown in **Table R11**, MEA fabricated Er SAC shows high full-cell energy efficiency of $\sim 34.7\%$ even at current density of 200 mA cm^{-2} .

Table R11. The full-cell energy efficiency (EE) of CO_2 to CO for Er SAC in MEA with acidic electrolyte.

Acidic electrolyte (MEA)			
Current density (mA cm^{-2})	Voltage (V)	FE_{CO} (%)	EE (%)
50	3.08	99.3	43.2
100	3.43	99.1	38.7
200	3.76	97.3	34.7
300	3.98	96.4	32.5
400	4.17	93.1	29.9
500	4.42	90.2	27.3

Reviewer #4 (Remarks to the Author):

The authors claim they developed a novel strategy utilizing atoms from the entire lanthanide group to facilitate the CO₂RR. The representative Er single atom catalyst exhibits a high Faradaic efficiency of CO exceeding 90%, a remarkable 42.2% full-cell energy efficiency and 70.4% single-pass CO₂ conversion efficiency. These results are significant. This paper is well organized, and the mechanisms are successfully demonstrated by authors. This work can be published in Nature Communications. There are some minor problems or issues that should be considered as follows.

1. Why did the author design the Er SAC with a coordination number of 6 in the DFT section. As know, the coordination number is relative to the performance of SAC. So, how about other coordination numbers for Er SAC, such as 7. Also, the author chooses the main group Ca as the center of SAC. I noticed that the other main group Mg atom can be used in CO₂ reduction. The author should assess the Mg SAC performance in the DFT prediction.

Response: Thank you for your helpful comment. We have designed the Er SAC model with a coordination of 6 mainly due to being consistent with the actual coordination number of synthesized Er SAC in the experiment. As expect, Fourier transformed (FT) extended X-ray adsorption fine structure (EXAFS) manifests the atomic dispersion feature of Er atoms, with a coordination number of ~6 based on well-fitting process respectively, consistent with the results from theoretical calculations.

As you suggest, we have assessed the Er SAC with the coordination number of 7 (Er-7 SAC) for predicting the CO₂RR performance with DFT calculations method. As exhibited in **Figure R28**, Er-7 SAC (1.08 eV) also exhibits a lower energy barrier for CO₂ activation than that of Ca SAC (1.46 eV), further proving the effectiveness of bridge adsorption strategy.

Figure R28. Free energy diagram of CO₂ reduction to CO for different catalysts.

To predict the Mg SAC, we have established the Mg SAC model with a 4 N coordination number (**Figure R29a**). As shown in **Figure R29b**, Mg SAC (1.75 eV) exhibits a much higher energy barrier of *COOH formation than that of Er SAC (0.61 eV), further verifying the effectiveness of bridge adsorption of *COOH.

Figure R29. a The model of Mg SAC. **b** Free energy diagram of CO₂ reduction to CO for Mg SAC and Er SAC.

2. The SAC with substrate of KB is synthesized in this work. The author should also provide the detail of preparation and suitable characterization results, such as TEM and SEM images.

Figure R30. **a** SEM, **b** TEM, **c** EDS mapping images for Er SAC with the substrate of KB.

Response: Thank you for your constructive suggestion. As shown in **Figure R30**, we have provided the detail of preparation and suitable characterization results for the Er SAC with substrate of KB.

3. Er SAC exhibits ~ 100 hours stability at 100 mA cm^{-2} . The authors should give some considerable characterization to analyze the state of catalyst after stability test and confirm the reason the performance decrease.

Response: Thank you for the constructive comment. As you suggest, we have conducted the XRD patterns, dark-field transmission electron microscopy (TEM) and EDS mapping for the Er SAC after stability test. As shown in **Figure R31**, no aggregation is found, and Er single atoms still exist in the CNT substrate. The reason for performance declining in flow cell is mainly due to the infiltration of gas diffusion layer (GDE) by electrolyte flood. As known, the stability of catalyst is highly relative to the hydrophobicity of gas diffusion layer. The CO_2 diffusion in flow cell will be intercepted by the electrolyte flood due to its decreased hydrophobicity during a successive electrocatalysis (**Figure R31a**).

Figure R31. **a** Gas diffusion electrode flooding after long-term operation in flow cell. **b** XRD pattern of Er SAC after stability test. **c** Dark-field scanning transmission electron microscopy and EDS mapping image of Er SAC after stability test.

4. Other paper has reported the potential of different primary products (CH_4 , CH_3OH) for Ln-based SACs (Langmuir 2023, 39, 41, 14748–14757). The author needs to check the NMR results about more than one SAC, such as Ce and Yb SAC.

Response: Thank you for the suggestion. We have checked the NMR results of Ce and Yb SAC. No C₂ and liquid products (such as ethanol, methanol) were observed by ¹H NMR, consistent with the result of Er SAC (Figure R32).

Figure R32. ¹HNMR spectra for Ce and Yb SAC after CO₂ reduction electrolysis at different potential in CO₂ saturated 0.5 M KHCO₃ solution.

Manuscript number: NCOMMS-24-64828A

Title: Lanthanide Single-Atom Catalysts for Efficient CO₂-to-CO Electroreduction

A Point-to-Point Response to Reviewer's comments

Dear Editor and Reviewers,

Thank you for taking the time and effort to carefully examine our manuscript. The comments are highly appreciated and helpful to improve this work. We have made corresponding changes to the manuscript and supporting information (highlighted in yellow in the revised versions) to address the editor's concerns and the requests from the reviewers; these changes are specified and discussed in a point-to-point response to the editor's and reviewers' comments, as shown below.

REVIEWER COMMENTS

Reviewer #1 (Remarks to the Author):

The authors have improved the manuscript. The proposed bridge adsorbed *COOH configuration is still not that convincing, in my opinion.

1. For my previous comment 2#, I was curious about the transition process from bridge adsorbed *COOH to linearly adsorbed *CO.

Response: Thanks for your comments. As shown in Figure R1, we have provided the total pathway of CO₂ reduction to CO on Er SAC. In step 2, the bridge-adsorbed *COOH converts into bridge-adsorbed *CO after proton coupled electron transfer (PCET) process. In step 3, the bridge-adsorbed *CO is very easy to spontaneously turn into linear-adsorbed *CO due to its high instability.

To verify this result, we initially assumed bridge-adsorbed *CO on Er site in DFT calculations. After energy optimization, we found the bridge-adsorbed *CO spontaneously turns into linear-adsorbed *CO, demonstrating the bridge-adsorbed *CO is hugely hard to exist stably.

We thought this result can be well-explained by the “back-donation” mechanism of *CO adsorption (*J. Phys. Chem.* **68**, 2772; *J. Am. Chem. Soc.* **1985**, 107, 578). The adsorption of *CO on metal sites always starts with the hybridization between 5σ orbit of *CO exhibits and metal atom. As exhibited in Figure R2, the direction of 5σ (*C-O) orbit should be perpendicular to catalyst surface, not allowing the formation of bridge-adsorbed *CO. Based on “back-donation” mechanism, the bridge-adsorbed *CO is highly unstable and thus easily convert into linear-adsorbed *CO.

Therefore, neglecting the highly unstable bridge-adsorbed *CO, we select the pathway with relatively stable intermediates in the manuscript.

Figure R1. Structure and adsorption configurations of key intermediates on Er SAC.

Figure R2. Schematic diagram of orbit hybridization between transition metal and CO 5σ orbit.

2. In the revised version, the authors report 70.4% single-pass CO₂ conversion efficiency at 200 mA cm⁻² with acidic electrolyte, which was obtained with 1 sccm CO₂ flow rate, as shown in Supplementary Table 6. Assuming a 100% CO Faradaic efficiency, the electroreduction of 0.7 sccm (1*70.4%) CO₂ to CO can only contribute a current of ~95 mA. Please double check this data. Can a high CO faradaic efficiency still be obtained at such a high CO₂ conversion efficiency with 1 sccm flow rate? Again, the full performance data set with Faradaic efficiencies at different flow rates should be

provided. The specific flow rates for various cell measurements should be clearly provided in the experimental sections.

Response: We highly appreciate that the reviewer gave very constructive comments on our studies. Following your suggestion, we have carefully double-checked the value of SPCE. We are sorry to only provide the values of faradaic efficiencies at different flow rates in Figure 4c, not in Supplementary Table 6.

As shown in Figure R3 (Figure 4c in the manuscript), the value of faradaic efficiency at 200 mA cm⁻² with 1 sccm in acidic electrolyte is 94.13%, not the assumed value of 100%. After calculations based on the equations (4) and (5), the Er SAC could reach 70.4% single-pass CO₂ conversion efficiency at 200 mA cm⁻² with 1 sccm CO₂ flow rate in acidic electrolyte. To directly show the data, we have provided the values of faradaic efficiencies at different flow rates in Supplementary Table 6.

The high CO faradaic efficiency can be obtained at 1 sccm flow rate and many works also achieved this goal, such as *Nat. Synth.* **2024**, 3, 1231. Indeed, some details should be noticed during the cell measurements and products analysis at low flow rates. Most noticeably, it took more than one hour and two hours of electroreduction time to wait for the gas components balance in the chamber when testing the faradaic efficiency at 2 and 1 sccm CO₂ flow rates respectively. Moreover, the GC standard curve was calibrated by using standard mixture gas with different concentrations, especially at high-concentration products at low flow rates.

Following your suggestion, we have provided cell measurements and products analysis in the experimental sections, especially at low flow rates.

Figure R3. FE_{CO} and current densities for Er SAC at different CO₂ flow rates at 1 M KCl (pH=1)

$$\text{CO}_2 \text{ consumed (L min}^{-1}\text{)} = (j \text{ mA cm}^{-2}) \left(\frac{1 \text{ A}}{1000 \text{ mA}} \right) \times \left(\frac{60 \text{ s}}{1 \text{ min}} \right) \times \left(\frac{1 \text{ mol e}^-}{96485 \text{ C}} \right) \times \left(\frac{1 \text{ mol CO}}{2 \text{ mol e}^-} \right) \times \left(\frac{1 \text{ mol CO}_2}{1 \text{ mol CO}} \right) \times \left(\frac{24.05 \text{ L}}{1 \text{ mol CO}_2} \right) \times (0.5 \text{ cm}^2) \quad (4)$$

SPC

$$(\%) = 100\% \times \left(\frac{\text{CO}_2 \text{ consumed (L min}^{-1}\text{)}}{\text{CO}_2 \text{ flow rate (L min}^{-1}\text{)}} \right) \quad (5)$$

Supplementary Table 6 | The single pass efficiency (SPCE) of CO₂ to CO for Er SAC at different CO₂ flow rates in acidic electrolyte (pH=1).

Flow rate (20 sccm)				Flow rate (10 sccm)			
j (mA)	FE _{CO} (%)	SPCE (%)	Error (% of SPCE)	j (mA)	FE _{CO} (%)	SPCE (%)	Error (% of SPCE)
100	99.21	1.7	3.5	100	91.98	3.4	2.3
200	99.72	3.7	3.7	200	98.29	7.4	1.6

Flow rate (5 sccm)				Flow rate (2 sccm)			
j (mA)	FE _{CO} (%)	SPCE (%)	Error (% of SPCE)	j (mA)	FE _{CO} (%)	SPCE (%)	Error (% of SPCE)
100	98.95	7.4	3.1	100	97.55	18.2	3.6
200	98.23	14.7	2.4	200	98.18	36.7	2.8

Flow rate (1 sccm)			
j (mA)	FE_{CO} (%)	SPCE (%)	Error (% of SPCE)
100	96.41	36.0	1.8
200	94.13	70.4	3.1

Reviewer #2

Reviewer #3

Reviewer #4

Response: We would like to thank the reviewers for the acceptance of our manuscript at this stage.

Manuscript number: NCOMMS-24-64828B

Title: Lanthanide Single-Atom Catalysts for Efficient CO₂-to-CO Electroreduction

A Point-to-Point Response to Reviewer's comments

Dear Editor and Reviewers,

Thank you for taking the time and effort to carefully examine our manuscript. We have made corresponding changes to the manuscript (marked in a yellow background) and supporting information to address the editor's concerns and the remaining requests of the reviewer; these changes are specified and discussed in a point-to-point response to the editor's and reviewer' comments, as shown below:

REVIEWER COMMENTS

Reviewer #1 (Remarks to the Author):

I find an important change of the electrode area in the equation (4), although the authors did not mention that in their rebuttal letter. The electrode area used in the equation is 0.5 cm² in the current version of the manuscript, but it was 1 cm² in the last two versions! That's why I have large doubts on the high single-pass CO₂ conversion efficiency of 70.4%. If 0.5 cm² were correct, it would be mathematically fine. In this regard, the values (100, 200) in Supplementary Table 6 should be current density with a unit of mA/cm², rather than current with a unit of mA. The expression “*j* (mA)” is also quite confusing, as *j* usually refers to current density. The electrode area and/or effective electrode area and electrolyte used in MEA cell should also be provided clearly. Is that also 0.5 cm²? The authors should be cautious and professional when presenting performance data.

Response: Thanks for your comment regarding this point. The actual effective electrode area in the flow cell is 0.5 cm². This value was always mentioned consistently across all previous versions in the Electrochemical measurements section: “the area contacting electrolyte is 0.5 cm².” While preparing the previous revision, we found that the electrode area value used in the Equation 4 was wrong (1 cm² instead of

0.5 cm²) and this is why we corrected it. We apologize for mistaken the correct electrode area in the Equation 4 in our original submission, but this was corrected already in the previous revision.

Moreover, following your suggestion, we have changed the expression “ j (mA)” to j (mA cm⁻²) in the revised Supplementary Table 6 (please see below).

Furthermore, as you suggest, we have provided clearly the effective electrode area and electrolyte of MEA cell in the Electrochemical measurements section and Supplementary Table 5. The effective electrode area in MEA is also 0.5 cm² (same as that in flow cell). The electrolyte in MEA is 0.5 M K₂SO₄ (pH adjusted to 1.0 with sulfuric acid), which has the same K⁺ concentration as the cathodic acidic electrolyte in flow cell, meanwhile avoiding chlorine evolution on the anode.

Besides this, to better present our performance data, we also carefully provided electrolyte details in the Electrochemical measurements section, Supplementary Fig. 37-40 and Table 4-6.

We hope that these clarifications are sufficient to complete the revision of our manuscript. We thank the reviewer for the fruitful feedback provided while revising our manuscript.

Supplementary Table 6 | The single pass efficiency (SPCE) of CO₂ to CO for Er SAC at different CO₂ flow rates with 1 M KCl (pH=1) electrolyte in flow cell.

Flow rate (20 sccm)				Flow rate (10 sccm)			
j (mA cm ⁻²)	FE _{CO} (%)	SPCE (%)	Error (% of SPCE)	j (mA cm ⁻²)	FE _{CO} (%)	SPCE (%)	Error (% of SPCE)
100	99.21	1.7	3.5	100	91.98	3.4	2.3
200	99.72	3.7	3.7	200	98.29	7.4	1.6

Flow rate (5 sccm)				Flow rate (2 sccm)			
j (mA cm ⁻²)	FE _{CO} (%)	SPCE (%)	Error (% of SPCE)	j (mA cm ⁻²)	FE _{CO} (%)	SPCE (%)	Error (% of SPCE)
100	98.95	7.4	3.1	100	97.55	18.2	3.6
200	98.23	14.7	2.4	200	98.18	36.7	2.8

Flow rate (1 sccm)			
j (mA cm ⁻²)	FE _{CO} (%)	SPCE (%)	Error (% of SPCE)
100	96.41	36.0	1.8
200	94.13	70.4	3.1